# Coordinated evolution at amino acid sites of SARS-CoV-2 spike

**Alexey Dmitrievich Neverov[1,2]\*, Gennady Fedonin[2,3,4], Anfisa Popova[2], Daria Bykova[2,5], Georgii Bazykin[4,6]**

[1]HSE University, Moscow, Russian Federation; [2]Central Research Institute for Epidemiology, Moscow, Russian Federation; [3]Moscow Institute of Physics and Technology (National Research University), Moscow, Russian Federation; [4]Institute for Information Transmission Problems (Kharkevich Institute) of the Russian Academy of Sciences, Moscow, Russian Federation; [5]Lomonosov Moscow State University, Moscow, Russian Federation; [6]Skolkovo Institute of Science and Technology, Moscow, Russian Federation

**Abstract** SARS-CoV-2 has adapted in a stepwise manner, with multiple beneficial mutations accumulating in a rapid succession at origins of VOCs, and the reasons for this are unclear. Here, we searched for coordinated evolution of amino acid sites in the spike protein of SARS-CoV-2. Specifically, we searched for concordantly evolving site pairs (CSPs) for which changes at one site were rapidly followed by changes at the other site in the same lineage. We detected 46 sites which formed 45 CSP. Sites in CSP were closer to each other in the protein structure than random pairs, indicating that concordant evolution has a functional basis. Notably, site pairs carrying lineage defining mutations of the four VOCs that circulated before May 2021 are enriched in CSPs. For the Alpha VOC, the enrichment is detected even if Alpha sequences are removed from analysis, indicating that VOC origin could have been facilitated by positive epistasis. Additionally, we detected nine discordantly evolving pairs of sites where mutations at one site unexpectedly rarely occurred on the background of a specific allele at another site, for example on the background of wild-type D at site 614 (four pairs) or derived Y at site 501 (three pairs). Our findings hint that positive epistasis between accumulating mutations could have delayed the assembly of advantageous combinations of mutations comprising at least some of the VOCs.

\*For correspondence: neva_2000@mail.ru

**Competing interest:** The authors declare that no competing interests exist.

## Editor's evaluation

Neverov and colleagues analyze patterns of correlated changes of amino acids in the SARS-CoV-2 spike protein to identify networks of interacting positions using an improved version of the previously validated method. Identifying such patterns of co-evolution is important for a better understanding of spike-protein evolution. The evidence for the identified co-evolving pairs is convincing, though the degree of certainty varies among the different identified groups of potentially interacting positions.

## Introduction

Evolution of SARS-CoV-2 in human hosts before November 2020 was largely neutral, with little evidence for emergence of novel adaptation with the exception of fixation of D614G in the Spike protein (*Dearlove et al., 2020*; *MacLean et al., 2021*). However, since the end of 2020, evidence for adaptive viral evolution has started to accumulate, suggesting a change in the mode of evolution (*Martin et al., 2021*; *Rochman et al., 2021b*). The subsequent pandemic was characterized

by emergence of multiple concurrently circulating lineages with increased fitness compared to the ancestral variant, including Alpha (B.1.1.7), Beta (B.1.351), Gamma (P.1), and Delta (B.1.617.2). These lineages are typically characterized by high divergence, compared to cocirculating strains; divergence is often particularly pronounced at nonsynonymous sites, suggesting positive selection at origin of these variants. Some of these sites are evident of a change in selection regime at the origin of VOCs. For example, out of the 34 lineage-defining amino acid changes in the S-protein at the origin of the Omicron BA.1 sublineage (**Hodcroft, 2021**), eleven were characterized by strong purifying selection against changes of ancestral amino acids (**Martin et al., 2022**), suggesting that Omicron lineage-defining mutations at these sites were previously individually deleterious. In turn, the obvious high fitness of Omicron suggested that the origin of this variant has been characterized by a change in the selection regime at least at these sites. The reasons for this change are unclear. Several non-exclusive explanations were proposed, including a distinct mode of evolution at variant origin, for example, in an immunosuppressed individual (**Corey et al., 2021**; **Kupferschmidt, 2021**) or a different host species (**Wei et al., 2021**) and/or cascades of substitutions at positively epistatically interacting sites (**Moulana et al., 2022**). Here, we focus on the latter possibility.

Several previous studies have attempted to infer possible epistatic interactions between sites of SARS-CoV-2 genome from sequence data or experimentally. In an early study, Zeng et al. used direct coupling analysis (DCA) to search for epistasis in SARS-CoV-2 genome and reported several pairs of putatively interacting sites (**Zeng et al., 2020**). No pairwise interactions between sites of the S-protein were identified. For DCA to accurately detect interacting sites, the analyzed sequences need to be highly divergent (**Bisardi et al., 2022**). SARS-CoV-2 has a recent common ancestor, and divergence of its lineages is relatively low, limiting the applicability of DCA for this virus. Rodriguez-Rivas et al. applied DCA to homologous protein sequences from genomes of other coronaviruses and successfully predicted variability of SARS-CoV-2 protein sites, thus showing that knowledge of covariation between sites in related viruses is relevant for predicting evolution of new pathogens (**Rodriguez-Rivas et al., 2022**). In the study of Rochman et al., a method based on counting of mutations on the phylogeny was used to look for strongly associated mutation pairs (**Rochman et al., 2020**). They found intra- and intergenic epistasis between positively selected mutations in the nuclear localization signal (NLS) of the N-protein and RBD in the S-protein. In RBD, many of the detected epistatically interacting mutations were among the lineage signature mutations. Another proposed approach to study epistatic interaction was to estimate the fitness effects of mutations that arose on different backgrounds relative to their effects on the wild-type background (**Rochman et al., 2021a**). Using molecular dynamics, Rochman et. al. estimated effects of all individual non-synonymous mutations in the S-protein RBM on binding with host ACE2 receptor and on binding with neutralizing Ab (NAb). Effects of each mutation were estimated for the Wuhan ancestral background, Delta (452 R, 478 K), Gamma variants (417T, 484 K, 501Y), and Omicron (339D, 371 L, 373 P, 375 F, 417 N, 440 K, 446 S, 477 N, 478 K, 484 A, 493 R, 496 S, 498 R, 501Y, 505 H). On average, the epistatic effects of mutations weakly stabilized NAb binding for Delta and destabilized it for Gamma and Omicron variants relative to the ancestral background. The authors concluded that the Gamma and Omicron variants had a higher potential for emergence of immune escape mutations than Delta or Wuhan variants.

For some site pairs, epistasis had been demonstrated experimentally. For example, the Q498R mutation alone affected the affinity of Spike to ACE2 only slightly (**Zahradník et al., 2021**), but on the background of N501Y, the affinity of binding increased by a factor of 4–25 (**Starr et al., 2022a**; **Zahradník et al., 2021**), with both mutations together increasing the affinity by up to 387-fold compared to the wild type (**Starr et al., 2022a**). The very strong binding provided by the double mutant allows accumulation, at Omicron origin, of multiple immune escape mutations at other sites which by themselves destabilize ACE2 binding (**Moulana et al., 2022**; **Starr et al., 2022a**).

Here, we study the mutual distribution of spike mutations in SARS-CoV-2 phylogeny to infer the pairs of sites with evidence for concordant and discordant evolution, as manifested by the propensity of substitutions at these sites to occur rapidly one after the other (for concordant evolution), or to avoid each other (for discordant evolution). We detect 46 concordantly evolved sites combined into 13 coevolving clusters, and 12 discordantly evolved sites. Many of the concordantly evolved sites carry the characteristic mutations of VOC lineages, strongly arguing for the role of positive epistasis in VOC origin.

## Results

### Detecting interdependently evolving pairs of sites

To find coevolving site pairs, we modified our previously developed phylogenetic approach (*Kryazhimskiy et al., 2011*; *Neverov et al., 2021*; *Neverov et al., 2015*) to improve the accuracy of detecting concordantly evolving site pairs (see Materials and methods). Similarly to our previous work, as a measure of concordance of evolution at two sites, we used the epistatic statistics calculated as the weighted sum of consecutive pairs of mutations at these two sites on the phylogeny, where each mutation pair was taken with exponential penalty for the waiting time for the later mutation (*Kryazhimskiy et al., 2011*).

We need to introduce some definitions for further explanation. Hereafter, unless specified otherwise, we use the term 'mutation' for defining a triple of a site, ancestral and derived amino acids identifiers. Using ancestral state reconstruction, we are able to infer the order in which two specific mutations occurred in an evolving lineage. For a pair of consecutive mutations at two sites, we call the mutation that occurs first a leading mutation, and the mutation that follows it, a trailing mutation. For an ordered pair of sites, we call the first site in a pair the background site, and the second site in the pair, the foreground site. The epistatic statistic for an ordered pair of sites summarizes the weights of consecutive pairs of mutations at these sites, such that mutations at the background site are leading and mutations at the foreground site are trailing. The epistatic statistics for an unordered pair of sites is a sum of statistics of the two corresponding ordered pairs (*Neverov et al., 2021*).

We introduced two significant changes to the original method (*Kryazhimskiy et al., 2011*; *Neverov et al., 2021*) which improved the power to infer epistasis ('revised method', see next section). First, as in *Neverov et al., 2015*, we modified the null model used to calculate the significance of the epistatic statistics in permutations. While previously (*Kryazhimskiy et al., 2011*; *Neverov et al., 2021*) we permuted the positions of mutations on the tree branches at each site independently of other sites, here, we fixed the positions of mutations for the background site and permuted just the positions of mutations at the foreground site. This change allowed us to account for the possible effects of leading mutations on the topology of the phylogenetic tree; for example, an advantageous leading mutation could give rise to a prolific clade (*Neher, 2013*) which in turn would carry a large number of trailing mutations, artefactually inflating the epistatic statistic for this pair of sites even in the absence of epistatic interactions.

Second, while our previous work (*Kryazhimskiy et al., 2011*; *Neverov et al., 2021*) treated all substitutions at a site equally, we now distinguished between substitutions into different amino acids. Therefore, the revised epistatic statistic accounts for the preference of a specific mutation at the foreground site to follow a specific mutation at the background site. For this, in calculation of the epistatic statistic, we now additionally scored each pair of consecutive mutations by the fraction of times that the specific type of mutation at the foreground site followed the specific type of mutations at the background site, among all occurrences of this type of mutations at the foreground site. Therefore, extra weight was given to those mutation types that became more frequent on a specific background.

### Estimating the power of the method to detect epistasis

To demonstrate that our revised method improves inference of positive and negative epistasis, we used MimicrEE2 (*Vlachos and Kofler, 2018*) to simulate clonal evolution of linked sites. We simulated two modes of evolution: (i) under positive and negative selection without epistatic interactions ('multiplicative mode'), and (ii) under epistatic selection ('epistatic mode').

Specifically, we simulated independent forward-time evolution of a population of 50,000 genotypes consisting of 100 biallelic sites. For the multiplicative mode, at the start of the simulations, 20 sites of the gene carried the disfavored allele, and were therefore under positive selection; and 20 sites carried the favored allele, and were therefore under negative selection. For the epistatic mode, 20 sites constituted 10 site pairs such that the sites within each pair evolved under positive epistasis; and another 20 sites constituted 10 site pairs such that the sites within each pair evolved under negative epistasis. Under each mode, the remaining 60 sites evolved neutrally (see Materials and methods).

We used simulations to estimate how the changes to the method for inference of epistasis introduced in this work impacted method accuracy. For this, using simulated datasets, we compared the power of the four variants of our method, corresponding to the presence or absence of the two

modifications introduced in the previous section (accounting for amino acid identities and unlinking the distributions of mutations on the tree branches for background and foreground for the null model).

To compare the specificity of the four variants of the method, we used the multiplicative mode of simulation (i.e. the absence of the epistasis), and asked how frequently concordant or discordant pairs were inferred under each model. Since there was no epistasis in the simulation, each such pair was spurious, and the best method would be the one with fewest such pairs. For each method, we counted the number of spuriously inferred concordant and discordant pairs at the lowest p-value threshold in our simulation trials (10⁻⁴). There were 24 concordant pairs and 16 discordant pairs in the method of *Neverov et al., 2021*, but just 2 concordant pairs and 2 discordant pairs in the revised method, indicating that the modifications introduced here helped improve the specificity of our approach. We used the 10% FDR level for this analysis and for all its variants (see below). For the FDR 10%, no concordant or discordant pairs were inferred in this dataset by the revised method ((*Appendix 1—tables 1 and 2*, *Appendix 1—figure 1*).

To study the accuracy of the four variants of the method, we used simulations with epistasis. The revised method detected all 10 positively epistatic site pairs as concordantly evolving; additionally, it spuriously detected five other site pairs as concordantly evolving (*Appendix 1—tables 3 and 4*). The revised method also detected 7 out of 10 negatively epistatic site pairs as discordantly evolving, and spuriously detected four other site pairs as discordantly evolving (*Appendix 1—tables 5 and 6*). The three other detection models were less accurate: the number of false predictions was greater than the number of true predictions for positive epistatic pairs for all other detection methods, and for negative epistatic pairs, for two out of three methods (*Appendix 1—tables 3 and 5*). Therefore, the revised method was the method of choice for subsequent analyses.

## Phylogenetic analysis of SARS-CoV-2 spike

To obtain a phylogeny representative of SARS-CoV-2 diversity, we downloaded 3,299,439 complete genome sequences of SARS-CoV-2 aligned to the WIV04 reference genome from the GISAID EpiCov database on 07.09.2021. We ignored insertions and deletions relative to the reference sequence and removed sequences with inframe stop codons in the spike protein. We then clustered the remaining sequences by pairwise distances between S-protein subsequences, allowing up to three mutations in the S-protein within a cluster, which resulted in 7,348 clusters. For each cluster, we selected one representative sequence of the best quality with the earliest date of sampling. The median date of representative sequences was February 10, 2021. Therefore, the dataset covered approximately equally both characteristic periods of SARS-CoV-2 evolution: the neutral period between Jan and Nov 2020, and the period of antigenic drift between Dec 2020 and May 2021 (*MacLean et al., 2021*; *Martin et al., 2021*). We classified representative sequences according to pangolin lineages. Most sequences (5,721) were of the B.1.* sublineages. The representative sequences included some from the variants of concern (VOCs) Alpha (B.1.1.7+Q.*, 951 sequences), Beta (B.1.351.*, 192 sequences), Delta (B.1.617.2+AY.*, 24 sequences) and Gamma (P.1.*, 100 sequences). The phylogeny of representative sequences was reconstructed using IQ-TREE (*Minh et al., 2020*). The tree was rooted by the outgroup USA-WA1/2020 (EPI_ISL_404895) that matched the sequence of the putative SARS-CoV-2 progenitor (*Bloom, 2021*; *Kumar et al., 2021*). The ancestral sequences at internal tree nodes were reconstructed by TreeTime (*Sagulenko et al., 2018*). We extracted the part of the alignment that corresponded to the S gene, and collapsed the internal tree branches without mutations in the S gene. The final tree had 1,783 internal branches. For each internal branch, we listed the amino acid mutations that occurred at this branch.

## Concordantly evolving site pairs

To study the concordant and discordant evolution of pairs of sites in SARS-CoV-2 spike, we applied our approach to the distribution of mutations in the S gene on the reconstructed SARS-CoV-2 phylogeny. 185 of the sites carried two or more mutations on internal tree branches. We considered all 17,020 unordered pairs of these sites.

We detected 45 concordantly evolving site pairs which comprised 46 sites (*Figure 1A*, *Appendix 1—table 7*, *Appendix 1—figure 2A*). Our phylogenetic approach for detecting concordantly and discordantly evolved site pairs relied on the assumption that the tree provided for analysis is correct. To check the robustness of our results to uncertainty of phylogenetic reconstruction, we repeated the

**Figure 1.** Concordantly evolving sites in SARS-CoV-2 Spike protein. (**A**) Clusters of concordantly evolving sites. Graph vertices represent sites, and edges represent concordantly evolving pairs (**Appendix 1—table 7**). The graph consists of 13 connected components, 8 of which contain just a single edge. Site pairs that were among the set of the best scoring pairs predicted for the alternative UShER (**Turakhia et al., 2021**) topology (FDR <10%, **Table 1**) are marked by asterisks. (**B**) Concordantly evolving sites among the lineage-defining sites of Alpha, Beta, Gamma, Delta (B.1.617.2+AY.*) and Omicron (BA.1) VOCs (**Hodcroft, 2021**). Concordantly evolving sites are colored in accordance with the clusters in panel A. Sites with fewer than 2 mutations which were not included in the analysis are in gray.

analysis on the tree reconstructed for the same set of sequences by the UShER (**Turakhia et al., 2021**) method utilizing maximum parsimony approach (see Materials and methods). Among the 45 site pairs inferred to be concordantly evolving (**Appendix 1—table 7**), 33 were also concordantly evolving on the UShER tree at 50% FDR, including 28 at 10% FDR (**Table 1**, **Figure 1A**, **Appendix 1—table 8**). Thus, we conclude that for 73% (33/45) of the detected concordantly evolved site pairs, the statistical signal was strong enough to be insensitive to phylogenetic uncertainty. In what follows, we focus on the IQ-TREE results.

To characterize the detected concordantly evolving site pairs, we considered the positions of their sites on the S-protein trimer structure (PDB ID: 7JJJ). The mean distance between coevolving site

**Table 1.** Concordantly evolving sites of the SARS-CoV-2 S-protein with FDR less than 10% for both reconstructed phylogenies (see Text). The following characteristics are shown: coordinates on the S-protein sequence, nominal p-values, the value of the epistatic statistics, the total number of consecutive mutation pairs for the two corresponding ordered site pairs, numbers of mutations in consecutive pairs at sites 1 and 2, total numbers of mutations at sites 1 and 2, and the distance in the protein structure (PDB ID: 7JJJ). Pairs of sites where non-consecutive mutations are further from each other than expected (suggesting both epistatic and episodic selection; p-value <0.05 after adjustment) are indicated in bold; pairs of sites where they are closer to each other than expected (suggesting episodic rather than epistatic selection) are indicated in italic (see *Appendix 1—tables 10 and 12*). Physical distance could not be calculated for site pairs (13, 152) and (681,716) because sites 13 and 681 were absent in 7JJJ.

| site 1 | site 2 | cluster | p-value | epistatic statistics | #consec. pairs of mutations | #mutations in consec. pairs at site 1 | #mutations in consec. pairs at site 2 | #mutations at site 1 | #mutations at site2 | physical distance, Å |
|---|---|---|---|---|---|---|---|---|---|---|
| 13 | 152 | | <2e-5 | 2.864 | 5 | 5 | 4 | 5 | 16 | - |
| 20 | 417 | 4 | 2.2e-4 | 1.348 | 4 | 3 | 2 | 8 | 7 | 47.83 |
| 26 | 190 | 4 | 1.8e-4 | 1.029 | 4 | 4 | 3 | 12 | 3 | 18.4 |
| 63 | 213 | 3 | 4e-5 | 0.681 | 1 | 1 | 1 | 2 | 3 | 13.51 |
| **69** | **70** | **3** | **<2e-5** | **1.125** | **2** | **2** | **2** | **5** | **4** | **1.3** |
| **70** | **144** | **3** | **<2e-5** | **1.301** | **3** | **3** | **3** | **4** | **5** | **14.03** |
| 76 | 490 | | 2.6e-4 | 0.22 | 1 | 1 | 1 | 3 | 3 | 45.18 |
| 189 | 360 | 2 | 1.6e-4 | 0.544 | 2 | 1 | 2 | 3 | 3 | 29.89 |
| 190 | 417 | 4 | 0.0001 | 1.272 | 3 | 2 | 2 | 3 | 7 | 39.93 |
| 259 | 261 | 3 | <2e-5 | 1.35 | *1.5* | 2 | 1 | 4 | 2 | 3.81 |
| 356 | 360 | 2 | <2e-5 | 1.283 | *2.5* | 2 | 3 | 2 | 3 | 10.12 |
| 359 | 360 | 2 | <2e-5 | 0.976 | *1.5* | 1 | 2 | 2 | 3 | 1.34 |
| 439 | 441 | 1 | <2e-5 | 2.097 | 4 | 3 | 4 | 9 | 8 | 3.5 |
| 440 | 441 | 1 | 1.4e-4 | 1.505 | 3 | 3 | 3 | 12 | 8 | 1.33 |
| 440 | 442 | 1 | <2e-5 | 2.476 | 3 | 2 | 3 | 12 | 6 | 3.92 |
| 440 | 444 | 1 | <2e-5 | 2.515 | *3.5* | 2 | 5 | 12 | 7 | 5.6 |
| 441 | 442 | 1 | <2e-5 | 2.244 | 4 | 4 | 4 | 8 | 6 | 1.33 |

*Table 1 continued on next page*

*Table 1 continued*

| site 1 | site 2 | cluster | p-value | epistatic statistics | #consec. pairs of mutations | #mutations in consec. pairs at site 1 | #mutations in consec. pairs at site 2 | #mutations at site 1 | #mutations at site2 | physical distance, Å |
|--------|--------|---------|---------|---------------------|----------------------------|--------------------------------------|--------------------------------------|---------------------|--------------------|---------------------|
| 441 | 443 | 1 | 2e-5 | 1.281 | 5 | 4 | 2 | 8 | 2 | 3.23 |
| 441 | 444 | 1 | <2e-5 | 3.492 | 5.5 | 4 | 4 | 8 | 7 | 2.6 |
| 442 | 443 | 1 | 6e-5 | 1.301 | 4 | 4 | 2 | 6 | 2 | 1.32 |
| 442 | 444 | 1 | <2e-5 | 3.013 | 6 | 4 | 4 | 6 | 7 | 4.05 |
| 443 | 444 | 1 | 8e-5 | 1.043 | 2 | 2 | 2 | 2 | 7 | 1.32 |
| 501 | 1118 | 5 | 1.4e-4 | 2.737 | 16 | 13 | 12 | 40 | 22 | 131.71 |
| 681 | 716 | 5 | <2e-5 | 3.905 | 16.5 | 12 | 16 | 59 | 21 | - |
| 716 | 982 | 5 | 2e-5 | 2.001 | 15 | 9 | 11 | 21 | 15 | 81.66 |
| 716 | 1118 | 5 | 4e-5 | 2.382 | 13 | 8 | 12 | 21 | 22 | 22.29 |
| 859 | 950 | | <2e-5 | 2.219 | 5.5 | 4 | 5 | 11 | 9 | 15.17 |
| 982 | 1118 | 5 | 1.4e-4 | 1.637 | 15 | 11 | 8 | 15 | 22 | 94.63 |

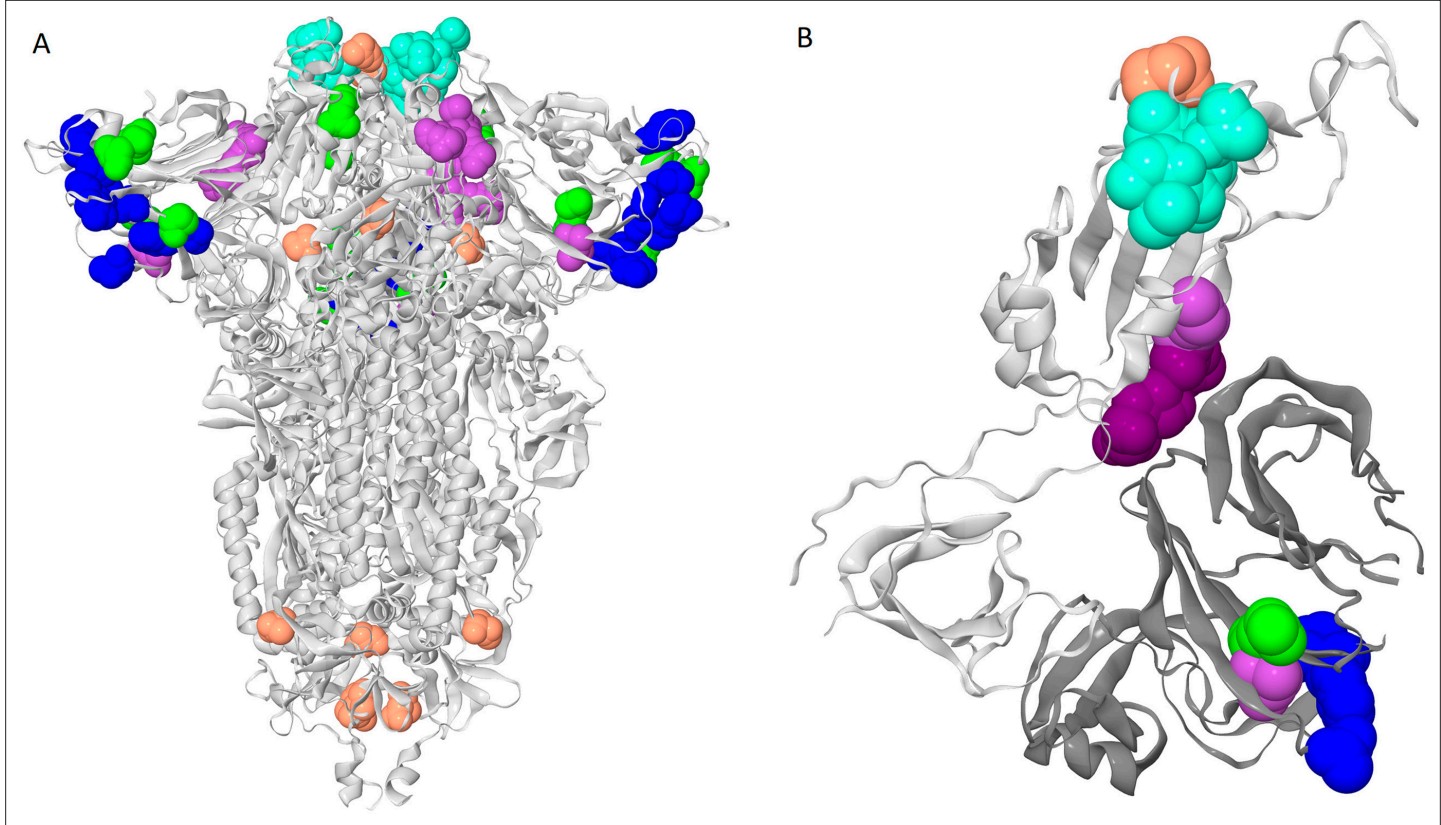

**Figure 2.** Clusters of coevolving sites on the protein structure. Sites of the five clusters that comprise multiple coevolving pairs of sites are shown as spheres, with color coding matching *Figure 1A*. (**A**) Closed conformation of the S-protein trimer (pdb: 7JJJ). (**B**) Open conformation of the S-protein trimer (pdb: 7KL9): for clarity, only residues 320–590 of one subunit and NTD 14–303 of the adjacent subunit are shown. NTD is shown in dark gray; residues 357, 359 and 360 are shown in dark purple.

pairs (20.17 A) was below that expected for random site pairs between the 185 sites with two or more mutations on internal branches (36.71 A, p=0.0004) as well as for random site pairs between the 46 sites involved in concordant evolution (36.64 A, p<1E-4). Among the 42 coevolving site pairs for which the distances on the protein structure were known, in 18 (43%), the two sites were in contact, that is, within 5 A from each other.

To visualize the detected concordantly evolving site pairs, we plotted them as a graph where vertices represented the 46 sites, and edges represented the 45 pairs formed by them (*Figure 1A*). The graph has 13 connected components; five of them contain between 5 and 9 sites, and the remaining 8 each consist of a single site pair. Sites of three of the five subgraphs with multiple vertices formed dense clusters on the protein structure, and for each such cluster, all or almost all sites belonged to the same domain; sites from the other two components were distributed dispersedly (*Figure 2A*). Hereafter, we referred to these clusters of sites by the Roman numerals I-V. All six sites of the first dense cluster 439–444 (I) were in the receptor binding motif (RBM) within the receptor binding domain (RBD). Mutations at sites of cluster (I) affect neutralization of SARS-CoV-2 by monoclonal and polyclonal antibodies (*Barnes et al., 2020*; *Harvey et al., 2021*). The four sites of the second dense cluster (II) 356, 357, 359, and 360 were in the RBD, while the fifth site 189 was in the NTD. Interestingly, in the open conformation (PDB ID: 7KL9), sites 357, 359, and 360 contacted the NTD domain of the adjacent subunit of the S-protein trimer, but in the closed conformation (PDB ID: 7JJJ) they were not in contacts (*Figure 2A and B*). The third dense cluster (III) 63, 64, 67, 69, 70, 144, 213, 259, and 261 was in the NTD domain in the region of binding of neutralizing antibodies (*McCarthy et al., 2021*). Four sites 18, 20, 26, and 190 in the first of the two dispersed clusters (IV) were in the NTD domain, and one site 417 was in the RDB. The second dispersed cluster (V) comprised sites from different domains: RBM (501), a position near the S1/S2 cleavage site (681), S2 (716 and 982) and the connecting domain (1118; *Figure 3*).

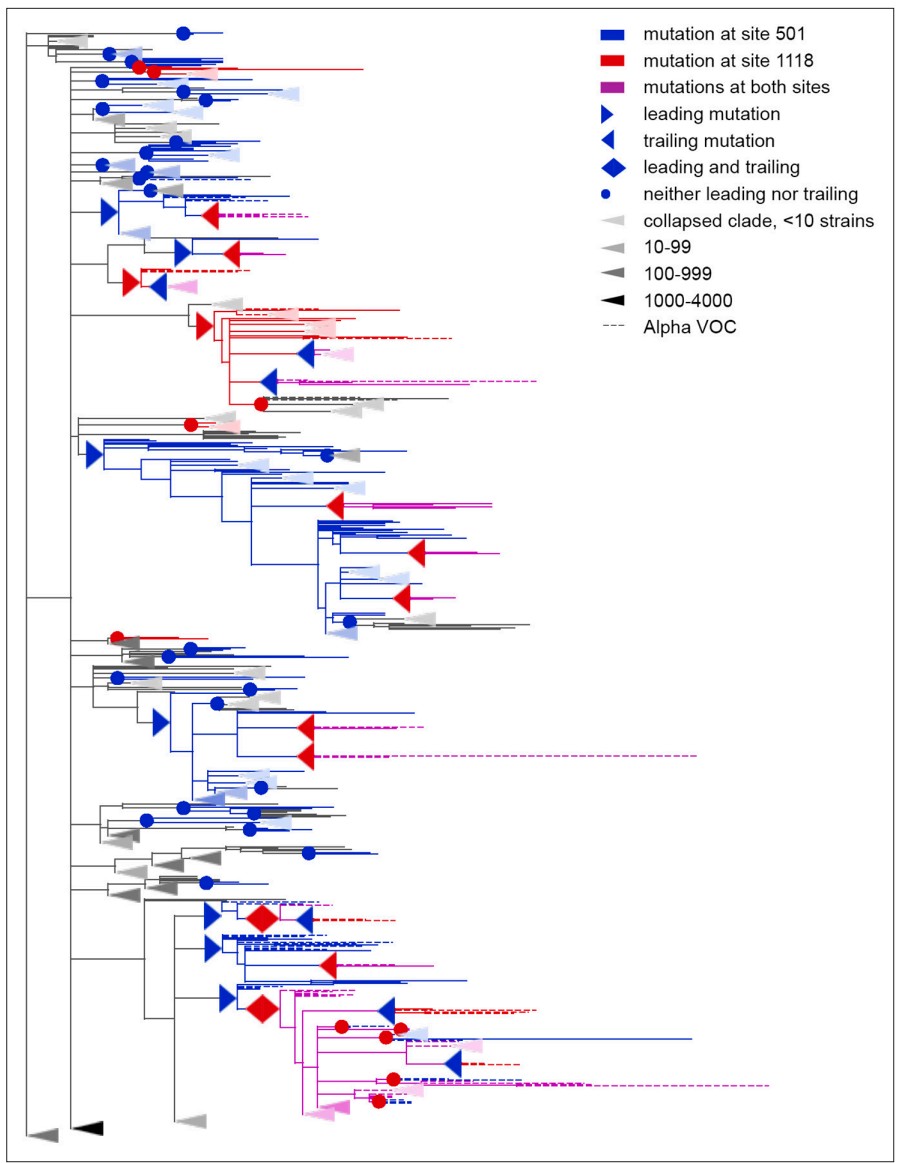

**Figure 3.** Concordantly evolving pair of sites 501 and 1118. Leading and trailing mutations are represented by blue (site 501) or red (site 1118) right-pointing and left-pointing triangles respectively; diamond-shaped signs indicate mutations that are both leading and trailing. All other mutations on internal branches, which are neither preceded nor followed by mutations at the other site on internal branches, are represented by circles. Mutations on terminal branches are excluded from the analysis and not shown (all mutations at these sites are shown on *Figure 3— figure supplement 1*). Branches carrying wild-type alleles (501 N and 1118D) are shown in black; those carrying substitutions at site 501, in blue; at site 1118, in red; at both sites, in violet. Here, leading and trailing mutations at site 501 are either N>Y or its reversion Y>N; and leading and trailing mutations at site 1118 are D>H or H>D. The dashed branches correspond to the Alpha VOC according to GISAID annotation as of 07.09.2021. For clarity of presentation, some of the clades without mutations at these two sites are collapsed and represented by elongate triangles, with intensity of color indicating the number of strains in the clade.

The online version of this article includes the following figure supplement(s) for figure 3:

**Figure supplement 1.** Concordantly evolving pair of sites 501 and 1118.

Many of concordantly evolving pairs of sites are located within loops in the NTD domain. Five out of the nine sites from the cluster (III) are located in the NTD hypervariable loops close to their flanks: sites 67, 69, and 70 are within the loop N2 (positions 67–81), site 144 is located within the loop N3 (positions 140–158), and sites 259 and 261 are within the loop N5 (positions 241–263). In the loop N1 (positions 14–26), there are three sites (18, 20 and 26) from the cluster (IV). Furthermore, the

**Table 2.** Discordantly evolving sites of the SARS-CoV-2 S-protein.

The following characteristics are shown: coordinates on the S-protein sequence, nominal p-values, the value of the epistatic statistic, the total number of consecutive mutation pairs for the two corresponding ordered site pairs, numbers of mutations in consecutive pairs at site 1 and site 2, total numbers of mutations at site 1 and site 2, FDR value corresponding to the p-value of the site pair obtained for the alternative phylogeny reconstructed by UShER (*Turakhia et al., 2021*), and the distance in the protein structure (PDB ID: 7JJJ). Pairs of sites where non-consecutive mutations are closer to each other than expected (suggesting both epistatic and episodic selection; p-value <0.05 after adjustment) are in bold; those where they are further from each other than expected (suggesting episodic rather than epistatic selection) are in italic (see *Appendix 1—tables 11 and 13*).

| site 1 | site 2 | p-value | epistat. | #consec. pairs of mutations | #mut. in consec. pairs in site1 | #mut. in consec. pairs in site2 | #mut. in site1 | #mut. in site2 | FDR UShER tree | physical distance,Å |
|---|---|---|---|---|---|---|---|---|---|---|
| 69 | 614 | 3e-3 | 0.07 | 5 | 1 | 5 | 5 | 14 | 0.231 | 46.75 |
| 222 | 501 | 3.1e-3 | 0.01 | 1 | 1 | 1 | 11 | 40 | 0.058 | 55.44 |
| **440** | **681** | **3.1e-3** | **0.01** | **1** | **1** | **1** | **12** | **59** | **0.055** | **-** |
| **501** | **675** | **1.2e-3** | **0.02** | **1** | **1** | **1** | **40** | **24** | **0.06** | **84.98** |
| 501 | 677 | 4.8e-4 | 0.05 | 3 | 2 | 3 | 40 | 39 | 0 | 88.20 |
| 570 | 614 | 2.4e-3 | 0.16 | 7 | 1 | 7 | 16 | 14 | 0.895 | 19.30 |
| *614* | *653* | *4e-5* | *0.03* | *4* | *1* | *4* | *14* | *5* | *0.025* | *15.47* |
| 614 | 982 | 3.2e-4 | 0.12 | 7 | 1 | 7 | 14 | 15 | 0.891 | 27.87 |
| 681 | 1176 | 3.9e-3 | 0.01 | 1 | 1 | 1 | 59 | 11 | 0.385 | - |

concordantly evolving site pair (155,157) comprises sites flanking the N3 loop. Yet another concordantly evolving pair (13, 152) comprises a site from the signal peptide and a site from the N3 loop. It has been previously shown that mutations at some sites of the signal peptide could abrogate virus neutralization by antibodies due to changes of the signal peptide cleavage site (*McCallum et al., 2021*). Changes of lengths and sequences of NTD loops mediate Spike membrane fusion, cell entry and extracellular stability (*Cantoni et al., 2022*; *Qing et al., 2021*). Loops N1 (positions 14–26), N3 and N5 contribute to NTD supersite of binding of neutralizing antibodies (*Cerutti et al., 2021*; *McCallum et al., 2021*).

## Discordantly evolving site pairs

We detected nine pairs composed of 12 sites that exhibited discordant patterns of evolution (*Table 2*, *Appendix 1—figure 2B*). Five of these site pairs were among the 16 discordantly evolving pairs between the 15 sites detected for the alternative UShER topology for the same level of FDR (*Appendix 1—table 9*). Similarly to concordant evolution, the signal of discordant evolution indicated that mutations at one site in a pair arose preferentially at specific allelic contexts of another site and avoided other contexts; e.g. mutations Q675H and Q677H occurred mostly on the wild-type background N in the 501 and rarely occurred on the background of 501Y (*Figure 4*), while mutations A653V and S982A occurred mostly on the derived background G at site 614 and rarely followed reversions to the wild-type D background.

Among the nine predicted discordantly evolving pairs, two, (501, 677) (shown on *Figure 4*) and (501, 675), are between the three sites whose effects of mutations were assessed experimentally or computationally. The N501Y mutation increases the binding affinity of Spike to ACE2 up to 15-fold (*Starr et al., 2022a*) and increases infectivity (*Liu et al., 2022*) the Q677H mutation increases infectivity, propensity to syncytium formation and escape of neutralization by serum of vaccinated people (*Zeng et al., 2021*) and the Q675H mutation is predicted to increase furin binding affinity (*Bertelli et al., 2021*). Although experiments suggested a positive effect of Q677H on the background of VOCs Alpha and Gamma both carrying N501Y (*Zeng et al., 2021*), this fact is in disagreement with the very low population frequencies of Q677H in these VOCs (*Gangavarapu et al., 2022a*, *Gangavarapu*

**Figure 4.** The Q677H is depleted on the background of N501Y. Branches carrying substitutions at site 501 are shown in blue; at site 677, in red; at both sites, in violet. There is less violet color than expected. Some clades without mutations at either site were truncated and are represented by elongate triangles, with color intensity indicating the number of strains in the clade.

*et al., 2022b*; *Gangavarapu, 2022c*; *Gangavarapu, 2022d*; *Khare et al., 2021*). The same is true for the Q675H mutation: while it was observed in some isolates of VOCs with the N501Y lineage-defining mutation, the population frequencies of these strains were also very low (*Bertelli et al., 2021*).

## Distinguishing between epistasis and non-epistatic episodic selection

The observed concordant and discordant evolution can stem from two sources: epistatic interactions between sites and episodic selection pressure affecting two or more sites at the same time. These two cases can be distinguished by patterns of phylogenetic distribution of non-consecutive mutations. Episodic selection simultaneously affecting two sites is expected to bias phylogenetic distances between all mutations at these sites, both consecutive and non-consecutive, so that substitutions are more likely to cooccur in more closely related lineages. By contrast, positive epistasis only leads to an excess of rapid consecutive mutations and does not bias distances between mutations in different lineages (*Neverov et al., 2021*). To check whether coordinated evolution of some pairs can be explained by concordant or discordant episodic selection alone, we applied the test described in *Neverov et al., 2021*, calculating the average distances between all pairs of non-consecutive substitutions for each concordantly or discordantly evolving pair of sites (*Appendix 1—tables 10–13*).

For the 12 concordant site pairs from 28 pairs that were detected on both phylogenies, we could not rule out that concordance stems from episodic selection alone, since their non-consecutive mutations are also evident of clustering (z-score <0, p-value after Benjamini-Hochberg adjustment <0.05). Specifically, for all concordantly evolving pairs within cluster I (sites 439–444, 10 pairs), non-consecutive mutations occur in more closely related lineages than expected, implying that their coordinated evolution could be a result of coincident episodic selection. Concordant evolution of the remaining 16 pairs cannot be explained without invoking positive epistasis, since their non-consecutive mutations, unlike consecutive ones, either tend to avoid each other [pairs (69, 70) and (70, 144) from cluster III, z-score >0, p-value after Benjamini-Hochberg adjustment <0.05] or at least do not show any signs of clustering (for 11 pairs, z-score >0; for the remaining 3 pairs, z-score <0 but p-value of clustering even before adjustment is greater than 0.15).

For the discordantly evolving pair (614, 653), non-consecutive mutations are more distant from each other than expected, suggesting that these pairs could also represent discordant episodic selection rather than epistatic interactions. By contrast, for discordant pairs (440, 681) and (501, 675) non-consecutive substitutions tend to be closer to each other than expected (in contrast to consecutive ones which repulse each other). This can only result from negative epistasis, probably accompanied by concordant episodic selection.

## Long-term coordinated evolution of Spike

We asked whether the concordantly evolving pairs of sites that we detect are also evident of long-term coordinated evolution on timescales of diversification within the larger group of sarbecoviruses. At such larger timescales, recombination, which is common between different sarbecoviruses (*Boni et al., 2020*; *Hu et al., 2017*; *Starr et al., 2022a*; *Wells et al., 2021*), becomes a major source of the evolutionary signal. Phylogenetic methods for inference of epistasis can be confounded by recombination (*Neverov et al., 2015*). Instead, we rely in this section on the DCA methods which are robust to moderate amounts of recombination (*Gao et al., 2019*). A previous study *Rodriguez-Rivas et al., 2022* used DCA to infer interactions within the five PFAM domains of Spike: bCoV_S1_N (PF16451.6), bCoV_S1_RBD (PF09408.11), CoV_S1_C (PF19209.1), bCoV_S2 (PF01601.17), and CoV_S2_C (PF19214.1). For each domain, we ordered site pairs by descending DCA scores, and calculated the ranks of concordantly and discordantly evolving pairs located within the corresponding domain. For each list, we referred to $N_D$ pairs with highest scores as 'high scoring pairs', where $N_D$ was the number of sites in the corresponding domain D, namely $N_D$ = 305 for bCoV_S1_N, 178 for bCoV_S1_RBD, 57 for CoV_S1_C, 519 for bCoV_S2 and 40 for CoV_S2_C.

Some of the concordantly evolving same-domain site pairs detected by us were also evolving in a coordinated fashion in sarbecoviruses in general. Specifically, 3 out of the 14 pairs of concordantly evolving sites in the bCoV_S1_RBD domain ((439, 441), (440, 442) and (441, 443), all from cluster I), 2 out of the 15 pairs of sites in the bCoV_S1_N domain ((63, 64) and (69, 70), both from cluster III) and one pair of sites in the Cov_S2_C domain (1258, 1259) were among the high scoring DCA pairs. The intersections of sets of concordantly evolving pairs of sites located within the same domain with

the sets of DCA high scoring pairs for corresponding domains were higher than expected by chance: p=4.7e-4 for bCoV_S1_RBD, p=3.2e-3 for bCoV_S1_N, and p=0.051 for CoV_S2_C (Fisher's exact test; the latter domain carried just one concordantly evolving pair). For the five domains with DCA results, no discordantly evolving pairs had both sites within the same domain.

## Coordinated evolution of sites carrying VOC mutations

Many of the concordantly evolving sites carried mutations that defined VOC lineages. In what follows, we discuss the sites of concordant mutations found among the sites carrying the characteristic mutations of specific VOCs (*Figure 1B*). We refer to sites bearing lineage-defining mutations as lineage defining sites.

For the Alpha VOC, 8 (69, 70, 144, 501, 681, 716, 982 и 1118) out of the 10 lineage defining sites (*Hodcroft, 2021*) were among the 46 concordantly evolving sites. Three of these sites, 69, 70 и 144, were within the dense cluster III of sites located in the NTD; the remaining five sites represented the dispersed cluster V.

For the Beta VOC, 3 sites (417, 501, and 484) out of the 10 lineage defining sites were among the concordantly evolving sites, but these sites did not form pairs with each other.

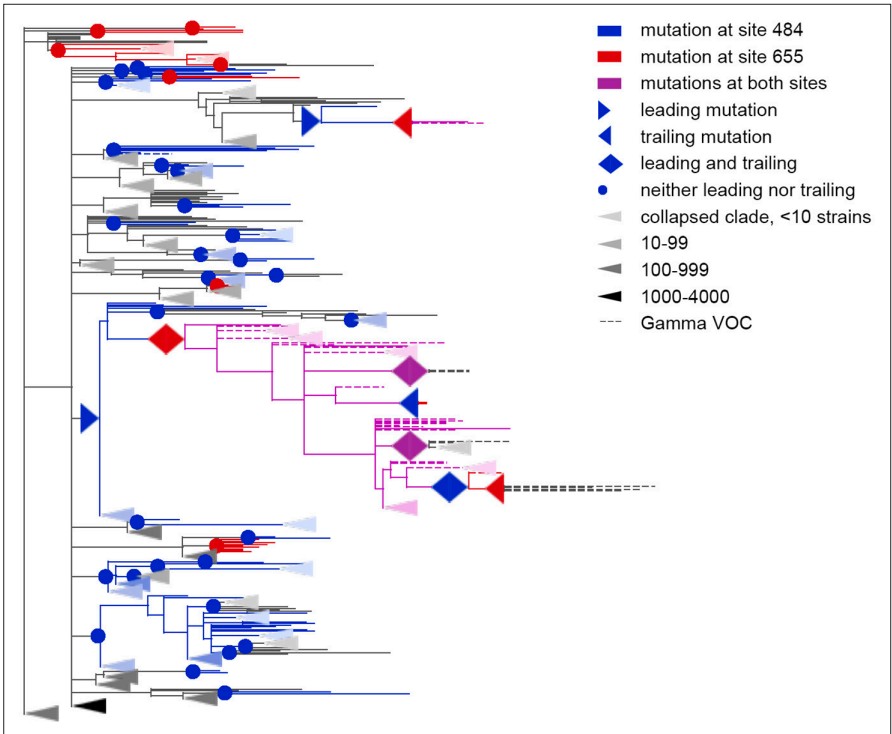

**Figure 5.** Coevolution of S-protein sites 484 and 655. Leading and trailing mutations are represented by blue (site 484), red (site 655) or violet (mutations at both sites on the same branch) right-pointing and left-pointing triangles respectively; diamond-shaped signs indicate mutations that are both leading and trailing, and violet signs indicate that both sites mutated on a single branch. All other mutations on internal branches, which are neither preceded nor followed by mutations at the other site on internal branches, are represented by circles. Mutations on terminal branches are excluded from the analysis and not shown (all mutations at these sites are shown on *Figure 5—figure supplement 1*). Branches carrying wild-type alleles (484E and 655 H) are shown in black; carrying substitutions at site 484, in blue; at site 655, in red; at both sites, in violet. Here, leading and trailing mutations at site 484 are either E>K or its reversions K>E; leading and trailing mutations at site 655 are H>Y or Y>H. The dashed branches correspond to the sequences of Gamma VOC according to GISAID annotation as of 07.09.2021. For clarity of presentation, some of the clades without mutations in these two sites are represented by elongate triangles, with color intensity indicating the number of strains in the clade.

The online version of this article includes the following figure supplement(s) for figure 5:

**Figure supplement 1.** Concordantly evolving pair of sites 484 and 655.

For the Gamma VOC, 7 out of the 12 lineage-defining sites (18, 20, 26, 417, 484, 501, and 655) were among the concordantly evolving sites. Four of these sites (18, 20, 26 and 417) were within the dispersed cluster IV of sites. Two sites (484, 655) constituted a distinct cluster with a single pair (*Figure 5*). Finally, site 501 had no concordantly evolving partners.

For the Delta VOC (B.1.617.2+AY.*), three out of the ten lineage defining sites (157, 681, and 950) were among the set of concordantly evolving sites, however, these sites belonged to different clusters and they did not form pairs with each other.

Finally, for the Omicron VOC (BA.1), 10 out of the 36 lineage defining sites (67, 69, 70, 144, 417, 440, 484, 501, 655, and 681) were among the concordantly evolving sites. Notably, no Omicron sequences were in the dataset used to assess concordance, so the signal observed at these sites is not due to clustering of substitutions in them at the origin of Omicron. Seven of these 10 sites, with the exception of the sites carrying deletions in Omicron (69, 70 and 144), were previously shown to evolve under positive selection (*Martin et al., 2022*). Among these 10 sites, four (67, 69, 70, and 144) were within the dense cluster III of sites located in the NTD. The sites 501 and 681 belonged to the highly dispersed cluster V of sites but did not form a pair, suggesting that any interactions between them could be indirect and mediated by other sites; site 501 is located within the RBM, while site 681 is near the S1/S2 furin cleavage site (FCS). Two sites 440 and 417 each were the only representatives of the corresponding cluster and had no concordantly evolving partners among the lineage-defining sites. However, site 440 was discordantly evolving with site 681 (*Table 2*). Mutations N440K and P681H are lineage-defining mutations of BA.1, but our analysis suggests that these mutations avoided each other before the appearance of Omicron due to negative epistasis. This indeed can be the case: despite the fact that the N440K mutation increases binding affinity to ACE2 and affects Ab neutralization efficiency (*Barnes et al., 2020*; *Harvey et al., 2021*; *Moulana et al., 2022*), by the end of 2021 its frequency in BA.1 was relatively low (<75%), while P681H reached 98% frequency (*Gangavarapu, 2022c*). However, the worldwide prevalence of the mutation pattern N440K+P681H+N501Y was increasing during the year and at the end of 2022, it was higher than 98% (*Gangavarapu et al., 2022e*). This may be explained by the acquisition of additional compensatory mutations (i.e. entrenchment of the N440K mutation on the background of P681H). To the best of our knowledge, there is no experimental confirmation of negative epistasis for these mutations; however, recent research by *Moulana et al., 2022* has shown that N440K exhibits weak negative epistasis with another Omicron lineage-defining mutation, N501Y. We also detected the discordant evolution of sites 440 and 501 for the USHER tree (*Appendix 1—table 9*). Together with the fact that 501 and 681 belong to the same cluster of concordantly evolving sites (cluster V), this supports the possibility of negative epistasis between sites 440 and 681. Finally, the two remaining sites, 484 and 655, formed a pair of coevolving sites which represented a separate single-edge cluster (*Figures 1A and 5*). Again, the first site in this pair was within the RBM and the second site was within the FCS. While 484 A is the characteristic allele of the Omicron VOC, it was rare in previously circulating strains: indeed, mutation E484A occurred in just a few (8 out of 7348) of the terminal branches of our phylogenetic tree and always in the context of the ancestral histidine at site 655. As our analysis disregards substitutions at terminal branches, E484A thus could not have contributed to our epistatic statistic. Instead, the signal of epistasis between sites 484 and 655 was formed by other mutations, notably, the frequently occurring E484K.

These findings suggest that the lineage-defining sites are enriched in concordantly evolving site pairs. To formally test this, we generated 400 artificial datasets by randomly redistributing the mutations on the phylogeny while preserving the numbers of substitutions at each site and on each branch of the tree, and applied our method for inference of coordinated evolution for each such dataset. For each pair of sites, we estimated the upper p-value as the probability to obtain the value of the epistatic statistic for independently evolving sites at least as high as that observed for the data. We ordered all site pairs in ascending order of their upper p-values, and compared the difference of mean ranks of pairs of lineage-defining sites and mean ranks of other pairs of sites, both for the real and for the random distributions of substitutions on the phylogeny (see Methods). All VOCs except Omicron had a stronger signal of coordinated evolution (lower ranks) for pairs of lineage-defining sites than for the remaining site pairs. This could not be explained by a difference in numbers of substitutions in lineage-defining and other sites because we have preserved numbers of substitutions at sites when generating datasets (*Appendix 1—table 14*; *Appendix 1—table 15*; *Appendix 1—table 16*;

*Appendix 1—table 17*; *Appendix 1—table 18*). Thus, our findings indicate that the lineage-defining sites of VOCs comprised pairs with an unexpectedly strong signal of concordant evolution.

As the lineage-defining sites of a VOC are by definition those that carry mutations at the origin of this VOC, the signal of concordant evolution at lineage-defining sites could arise from clustering of mutations at these sites at VOC origin. To address this confounding factor, we asked whether concordant evolution of lineage-defining sites of a VOC is also observed when this VOC itself is excluded from analysis. For this, we separately pruned all isolates belonging to each of the four VOCs Alpha, Beta, Gamma, and Delta from the tree. Note that our datasets did not contain Omicron from the start. For each of the four pruned trees, we separately predicted the concordantly evolving site pairs, and tested for enrichment of pairs of lineage-defining sites. For Alpha, pairs of lineage-defining sites still had a stronger signal of concordant evolution, indicating that at least for this VOC, the observed concordance is not due to clustering of substitutions at VOC origin (*Appendix 1—table 19*). For the other three VOCs, Beta, Delta, and Gamma, no enrichment was detected (*Appendix 1—table 20*; *Appendix 1—table 21*; *Appendix 1—table 22*). In fact, for Gamma, after pruning of the VOC clade, pairs of lineage-defining sites became depleted among the pairs with a stronger signal of concordant evolution (*P*=0.0272, *Appendix 1—table 22*); in this VOC, the observed signal of concordant evolution of lineage-defining sites is mainly caused by reversions of lineage-defining mutations within the VOC clade (*Figure 5*).

## Discussion

In this work, we modified the phylogenetic approach for detection of interdependently evolving pairs of sites that had been previously successfully applied for influenza A (*Kryazhimskiy et al., 2011*; *Neverov et al., 2015*) and mitochondrial proteins (*Neverov et al., 2021*), and applied it to the Spike protein of SARS-CoV-2. Using simulations, we show that our revised method has a better specificity and sensitivity for detection of epistatically interacting site pairs than its original version, and that it is able to detect positive as well as negative epistasis between alleles. Our simulations involved multiple sites with unfavorable alleles, so there were multiple adaptive mutations, allowing competition between multiple clones and hitchhiking. Despite these confounders, our method produced no spurious signal of epistasis when epistasis was not a part of the simulation (*Appendix 1—table 1* and *Appendix 1—table 2*, *Appendix 1—figure 1*) and was able to detect true interacting site pairs with a reasonable FDR for the epistatic mode of evolution of genotypes (*Appendix 1—table 3*; *Appendix 1—table 4*).

Similarly to other methods for inference of factors of evolution from comparative genomics data, the accuracy of our method depends on validity of its assumptions. First, we assume that the observed changes in the ancestral reconstructed states between adjacent nodes of the phylogeny are not artefactual but correspond to actual mutations. Unfortunately, for large-scale sequencing projects such as that of SARS-CoV-2, some extent of mistakes in the called sequences is unavoidable, and these mistakes are not random. Specifically, as noted previously (*Martin et al., 2022*), the propensity to call the ancestral nucleotides (reference bias), particularly at sites of low NGS read coverage, may lead to reversions to wild-type alleles, in particular for lineage-defining mutations. Indeed, the main cause of calling of wild-type alleles at multiple sites is the selective preference of PCR or sequencing primers to some genotypes that leads to a high variation in coverage along the genome of different isolates. In periods of change of the dominant variant, two genetically different variants circulate with high population frequencies, and mixed infections or cross-contaminations of samples may result in artifactual hybrid sequences. This can be a problem for our method. Exclusion of terminal tree branches mitigates these problems by focusing the analysis on more frequent genotypes because spurious reversions are more likely in individual sequences than larger clades. This problem is also partially addressed by the fact that we count all trailing mutations following a single leading mutation as one, so our statistic favors site pairs with multiple phylogenetically unrelated pairs of consecutive mutations; this requirement decreases the impact of a systematic loss of mutations at some sites of specific variants of genotypes.

Second, our approach relies on the correctness of the phylogeny. The inference of the true phylogeny for SARS-CoV-2 is difficult due to the huge amount of data and limited sequence diversity (*Morel et al., 2021*; *Rochman et al., 2021b*). To assess the impact of phylogenetic uncertainty on the results of analysis, we compared the sets of concordantly and discordantly evolving site pairs inferred

for two trees: the maximum likelihood (ML) tree obtained by IQ-TREE 2 (*Minh et al., 2020*) and maximum parsimony (MP) tree obtained by UShER (*Turakhia et al., 2021*). The choice of the method of phylogenetic reconstruction affected the set of predicted pairs (*Table 1*, *Appendix 1—table 7*; *Appendix 1—table 8*); however, over 60% of concordantly evolving pairs detected using the IQ-TREE phylogeny were also detected using the UShER tree.

Third, we assume that evolution is clonal, so that the phylogenetic tree reflects the true evolutionary history of the genome. If genotypes recombine, some genomic sites would be incompatible, that is, have different evolutionary histories (*Bruen et al., 2006*). On the whole genome phylogenetic tree, those sites with evolutionary history disagreeing with the majority would be enriched in spurious parallel or convergent substitutions, possibly affecting the signal of concordance. While the recombination frequency for SARS-CoV-2 is unknown, it has been estimated that about 3–5% of circulating genotypes are recombinants (*Kozlakidis, 2022*; *Turkahia et al., 2021*) and the frequency of occurrence of recombination breakpoints in the S gene is up to three times higher than in the rest of the genome (*Turkahia et al., 2021*). While our dataset included four sequences of the XB recombinant lineage, exclusion of these sequences did not affect our results. Nevertheless, some of the recombinants could remain unannotated, particularly those including just a few samples and/or originating from similar sequences.

The signals of concordant and discordant evolution that we observe could come from at least two natural sources. Firstly, they could arise from epistatic interactions between sites, an explanation which we favored in our previous works (*Kryazhimskiy et al., 2011*; *Neverov et al., 2021*; *Neverov et al., 2015*). Previously, *Ruan et al., 2022* interpreted the stepwise accumulation of mutations at origin of Gamma and Delta VOCs as evidence for epistatic interactions between lineage defining mutations. Under this explanation, concordant evolution of a pair of sites is indicative of positive epistasis between the newly arising alleles, and discordant evolution, of negative epistasis between them. Recently, epistasis in the Spike protein has been demonstrated experimentally: at 15 of the RBD sites, the effects of mutations were shown to change on the background of N501Y (*Starr et al., 2022a*). Unfortunately, it is hard to cross-validate the epistatic interactions between the evolutionary and experimental analyses. In our dataset, these sites are conserved: for 14 out of the 15 sites, there are no mutations on internal tree branches, making them unfit for our analysis. The remaining site 449 carried a single mutation Y449H which by itself is known to strongly decrease ACE2 binding affinity (*Starr et al., 2022a*) this mutation co-occurred with N501Y on one internal branch which had two descendent leaves, leading to a p-value of 0.18 for this pair in our test.

Several lines of evidence indicate that the observed coordination of evolution at different sites is largely due to epistatic, rather than (or at least in addition to) episodic, selection. First, over a half of concordantly evolving site pairs that we detected lack any signs of clustering of non-consecutive substitutions, which are expected if concordance results from coincident episodes of selection (*Neverov et al., 2021*), and thus can only be interpreted as epistatic pairs. The same applies to some discordantly evolving pairs, where non-consecutive mutations, unlike those occurring in the same lineage, do not show any signs of repulsion. A second piece of evidence comes from the observation of discordant evolution at site pairs (501, 675) and (501, 677) carrying mutations which are likely individually beneficial - a pattern expected under epistasis but not episodic selection. Third, six of the concordantly evolving pairs also experience coordination over the long term evolution of sabrecoviruses. The fitness landscape experienced by this group at large evolutionary timescales is likely different from the short-term landscape of SARS-CoV-2, e.g. because the Spike affinity to ACE2 could be completely lost in some viral lineages or easily expanded to new host species (*Starr et al., 2022b*). On such long-term time scales, the conservation of the protein fold is likely the strongest evolutionary constraint, and the fact the same site pairs evolve in a coordinated fashion within SARS-CoV-2 suggests that this short-term evolution is also shaped by epistasis.

The second natural source of coordinated occurrence of mutations is differences in selection regimes between tree branches. The observed phylogenetic clustering of non-consecutive mutations at concordant site pairs, and phylogenetic repulsion of non-consecutive mutations at discordant site pairs, imply that many concordantly and discordantly evolving site pairs were subject to coordinated episodes of adaptation, coincident or separate. When the pattern is the same for both consecutive and non-consecutive mutations, it is impossible to determine whether there is any epistatic interaction between sites, or the coordinated evolution results from coordinated additive selection alone. Over

the course of the pandemic, selection on the virus has changed due to the dynamics of herd immunity of the human population, with selection favoring immune escape mutations increasing with time. In particular, this was proposed to underlie the accelerated evolution and selective advantage of Gamma (*Gräf et al., 2021*, p. 1) and Omicron (*Martin et al., 2022*) VOCs. Additionally, distinct selection regimes could correspond to long-term infection in immunocompromised individuals or in non-human hosts under reverse zoonosis events.

Overall, we find that the VOCs at their origin have gained mutations at concordantly evolving sites (*Appendix 1—table 14*; *Appendix 1—table 15*; *Appendix 1—table 16*; *Appendix 1—table 17*; *Appendix 1—table 18*). Has this selection been episodic or epistatic? Both hypotheses have support. On the one hand, the set of the VOCs that we indicate as evolving in a coordinated manner, Alpha, Gamma, and Omicron, are the same three VOCs that are characterized by an increase in the rate of evolution at their origin and therefore have been suggested to had evolved in distinct evolutionary regimes, perhaps that of an immunocompromised patient (*Hill et al., 2022*; *Martin et al., 2022*). A possible explanation of constellations of lineage-defining mutation at the origin of VOCs is that selection in favor of these mutations increases in immunocompromised patients, leading to episodes of adaptation (*Martin et al., 2022*) during which the rate of their accumulation is increased compared to the baseline observed in the general population, making them clustered in the branches corresponding to such patients. However, it is unclear why the virus had to 'wait' for an immunocompromised individual to evolve the mutations that also additively increased its fitness in the general population. The late emergence of these constellations of mutations instead suggests that the selection had to be non-additive, that is, epistatic. Our observation that lineage-defining sites of Alpha evolve concordantly even outside the Alpha clade further supports the significance of epistasis at origin of at least this VOC (*Appendix 1—table 19*).

Finally, the signal of concordant evolution at the origin of VOCs could come from the combination of both factors: a distinct selection regime and epistasis. The role of immunocompromised patients in the evolution of SARS-CoV-2 is even higher for epistatic than for the non-epistatic mode of evolution (*Smith and Ashby, 2022*). If the viral fitness, as it has been previously proposed (*Hill et al., 2022*; *Rochman et al., 2021a*; *Smith and Ashby, 2022*), is a trade-off between transmission efficiency and ability to avoid herd immunity, and the transmission component of the fitness landscape has valleys of low-fitness genotypes, the immunocompromised individuals with prolonged infectious periods due to relaxed selection for transmission efficiency allow the virus to accumulate mutations and cross these valleys. Direct experimental studies of the possible epistatic interactions between coevolving sites will help elucidate the mechanism of origin of radically novel viral variants.

## Materials and methods
### Constructing the set of sequences
Masked full genome sequence alignment and corresponding metadata were downloaded from https://gisaid.org/ on 07.09.2021 comprising sequences for 3,299,439 isolates (see data availability statement). Based on metadata, all sequences from nonhuman hosts, without collection dates, or with wrongly formatted collection dates were excluded. For each sample, the number of gaps, 'N's and ambiguous characters were computed for each genome sequence and separately for the S-gene in non-masked positions. For each sequence, we calculated the number of positions that contained a nongap, non-'N' symbol that were aligned to non-gap positions of the reference genome sequence WIV04. We excluded sequences from the analysis with fewer 29,000 aligned positions, or with <95% of sequence length in aligned positions. Next, we sorted sequences by sampling dates and then converted each sequence into a list of changes relative to the reference, treating consecutive gaps as one change. Then, we excluded sequences having too many changes for their sampling dates. For that, for each sampling date, we computed the mean and standard deviations of the number of changes for all sequences whose sampling dates were within a half year time interval centered at this date. All samples with the number of changes exceeding the mean value by two standard deviations or more in the corresponding time interval centered on the sampling date were filtered out. Samples with preliminary stop codons inside the S gene were also excluded. This left us with 2,676,884 sequences for further analysis.

We partitioned the remaining sequences into groups of equivalence such that all sequences in each group had the same list of mutations in the S gene, and selected the sequence with the earliest collection date as the class representative. To further cluster the sequences, we reimplemented the UCLUST (*Edgar, 2010*) algorithm in a custom python script to allow it to process the huge amount of SARS-CoV-2 data. In contrast to the original UCLUST implementation that receives nucleotide or amino acid sequences as input, our script clustered the lists of changes in sequences that occurred relative to the sequence of the reference genome. This sped up computation, because there were few changes in SARS-CoV-2 sequences compared to the genome length. The pairwise distance between the two lists of changes was defined as the number of changes unique to each list. At the start of the procedure, samples were ordered by the sampling date; the first sample was defined to be the centroid of the first cluster and added to the list of centroids. Next, the remaining samples were iteratively compared with the centroids in the list: if the distance to some centroid was less than three, the sample was added to the corresponding cluster, otherwise it was added as a new entry to the list of centroids. Thus, by construction, cluster centroids tend to be the earliest representatives of their members. For each cluster, all samples from the corresponding groups of equivalent sequences were pooled together, and the sample having the highest quality sequence was selected as the representative of the cluster. The highest-quality sequence was defined as the sequence having the minimal number of gaps or 'N' characters in the S gene. If several sequences complied with the previous condition, the sequence with the minimal number of ambiguous characters in the S gene was selected as the representative of the cluster. If there were more than one such sequence, the one additionally having the minimal number of gaps or 'N' characters in the whole genome sequence was selected. If still more than one sequence met all the previous conditions, the first of them which had the minimal number of ambiguous characters in the whole genome sequence was finally chosen. All gaps in the selected complete genome sequences were converted to reference characters, and insertions relative to reference sequence were ignored.

## Phylogenetic analysis and inference of ancestral sequences

The phylogenetic tree was constructed with IQ-TREE (v. 2.1.2, model = GTR + I+G) (*Minh et al., 2020*). Additionally, we obtained an alternative topology for these sequences by UShER (*Turakhia et al., 2021*) for this, we inserted the sequences into a prebuilt global SARS-CoV-2 phylogeny of publicly available genome sequences (http://hgdownload.soe.ucsc.edu/goldenPath/wuhCor1/UShER_SARS-CoV-2/public-latest.all.masked.pb.gz accessed on 10.01.2022). Some of our selected GISAID sequences had already been in the global tree, so we inserted only those which were absent there. Finally, we extracted the subtree corresponding to the analyzed sequences from the global tree.

The trees were rooted by the outgroup sequence USA-WA1/2020 (EPI_ISL_404895).

For both trees, ancestral sequences were reconstructed by TreeTime (v. 0.8.2) (*Sagulenko et al., 2018*) with default parameters. Because we focused on the evolution of the S gene, we removed from each tree the internal nodes which had S-gene sequences identical to their parental nodes. Finally, we obtained the list of mutations for each tree branch.

## Concordantly and discordantly evolving pairs of sites

Our approach to detection of concordantly and discordantly evolving pairs of sites is a development of the phylogenetic method published earlier (*Kryazhimskiy et al., 2011*; *Neverov et al., 2021*; *Neverov et al., 2015*). It is based on counting consecutive pairs of mutations on the branches of a phylogenetic tree. A pair of mutations at two different sites is called consecutive if a mutation in one of the sites occurs in the subtree of the branch which carries a mutation in the other site, and there are no other mutations at these sites on the branches that constitute the path between them; if two sites mutate on the same branch, we assume that both orders are equiprobable (*Kryazhimskiy et al., 2011*). Here, we consider four models for detection of epistasis which are based on this approach. Two of these models account for identities of ancestral and derived amino acids at sites; the other two models disregard the identities of amino acids and account only for occurrences of mutations on the tree branches. We define the epistatic statistics in a general form which is used for all four models. The expression of the epistatic statistic for models that ignore identities of alleles could be straightforwardly obtained from the general form.

For an ordered pair of sites $(i, j)$, the epistatic statistics $e_{(i,j)}$ is the weighted number of consecutive pairs. It is formally defined as.

$$e_{(i,j)}\left(\alpha, A, \beta, B\right) = \sum_{l \in E} \frac{\delta_i\left(l, \alpha, A\right)}{\sum\limits_{m \in E(m \succcurlyeq l)} \delta_{i,j}\left(m \succcurlyeq l\right)} * \sum_{m \in E(m \succcurlyeq l)} \delta_{i,j}\left(m \succcurlyeq l\right) \delta_j\left(m, \beta, B\right) c_{(i,j)}\left(l, m\right) e^{-t_{l,m}/\tau}$$

$$w_{(i,j)}\left(\alpha, A, \beta, B\right) = \frac{\sum\limits_{l \in E} \delta_i\left(l, \alpha, A\right) \sum\limits_{m \in E(m \succcurlyeq l)} \delta_{i,j}\left(m \succcurlyeq l\right) \delta_j\left(m, \beta, B\right)}{\sum\limits_{m \in E} \delta_j\left(m, \beta, B\right)}$$

$$e_{(i,j)} = \sum_{(\beta, B)} \sum_{(\alpha, A)} e_{(i,j)}\left(\alpha, A, \beta, B\right) w_{(i,j)}\left(\alpha, A, \beta, B\right)$$

Here, $\alpha$ and $\beta$ are the ancestral alleles, and $A$ and $B$ are the derived alleles at sites $i$ and $j$; $E$ is the set of all tree edges; and $E\left(m \succcurlyeq l\right)$ is the set of edges descendant to the edge $l$. Indicator functions $\delta_i\left(l, \alpha, A\right)$ and $\delta_j\left(m, \beta, B\right)$ equal to one if on branches $l$ and $m$, mutations at sites $i$ and $j$ occur from the specified ancestral alleles to the specified derived alleles. The indicator function $\delta_{i,j}\left(m \succcurlyeq l\right)$ equals to one if a mutation occurs at branch $l$ at site $i$, and a consecutive mutation occurs at branch $m$ at site $j$. $t_{l,m}$ is the length of the shortest path between tree branches $l$ and $m$. The function $c_{(i,j)}\left(l, m\right)$ accounts for possible incomplete time resolution of the sequence of occurrence of mutations: it equals one if mutations at sites $i$ and $j$ occur on different branches, 0.5 if mutations at both sites occur on the same branch $l = m$, 1.5 if mutations at both sites occur at the same branch $l$ or $m$ and are followed or preceded by a mutation at one of the sites $i$ or $j$ at another branch, and 0.25 if mutations at both sites occur on both branches $l$ and $m$. The weight $w_{(i,j)}\left(\alpha, A, \beta, B\right)$ is the fraction of all mutations from $\beta$ to $B$ at site $j$ that occur on the background of the mutation from $\alpha$ to $A$ at site $i$. In the models of detection of epistasis which ignore allele identities, the weights $w_{(i,j)}\left(\alpha, A, \beta, B\right)$ are set equal to 1. The epistatic statistic for an unordered pair is a total of two statistics for ordered pairs: $e_{i,j} = e_{(i,j)} + e_{(j,i)}$.

We used two different null models for the epistatic statistics. The first model randomly and independently reshuffles mutations at each site; the second model specifically accounts for distribution of mutations at each ordered pair of sites $(i, j)$. To do that, the phylogenetic positions of mutations at the first site of a pair (background site) $i$ are considered fixed, and the positions of mutations at the second (foreground) site $j$ are reshuffled, thus unlinking the background and foreground. For both null models, reshuffling of mutation positions between tree branches preserves the total number of mutations at each site and the total number of mutations at each branch by using BiRewire utility (*Gobbi et al., 2014*). Combinations of two variants of epistatic statistics (with and without alleles) and two variants of the null model (with linked and with unlinked background and foreground) provide four models for detection of site pair epistasis which are compared in this study.

For the two models that account for allele identities, we generate amino acid sequences for internal and terminal nodes of the tree. For that, starting from the root sequence and traversing the tree from root to tips, we generate derived alleles for each mutation on a tree branch from the empirical allele distributions at this site, conditioned on the allele at the parental node of the branch. For the null model with unlinked background and foreground, the sequences in the tree nodes for the background remain unchanged, while for the foreground, new sequences are generated. For generating random distributions of mutations on tree branches and for calculation of epistatic statistics, we used the Bio::-Phylo Perl module (*Vos et al., 2011*) for traversing phylogenetic trees.

For each model, we perform 50,000 permutations of positions of mutations; for the two allele-aware models, for each permutation, we also generate the amino acid sequences at tree nodes. For each permutation, we calculate the epistatic statistics for unordered pairs, together with two p-values: the fraction of statistics equal or greater than the observed value (upper p-value), and the fraction of statistics equal or less than the observed value (lower p-value).

To account for multiple testing, we estimated the false discovery rates (FDR). For this, we randomly selected 400 out of 50,000 permutations. For each of the 400 permutations, for each unordered pair of sites, we calculated the epistatic statistic and the upper and the lower p-values. For each p-value threshold, we calculate the corresponding number of findings for the real dataset (R – declared positives) and the average number of findings in the fake dataset (E[V] – false positives). The FDR is the ratio of E[V] to R.

To differentiate between epistasis and non-epistatic episodic selection, for each concordantly or discordantly evolving pair of sites, we analyzed phylogenetic distances between nonconsecutive substitutions as described in *Neverov et al., 2021*, with the only difference that 400 sets of mutations instead of 200 were generated for the null model to obtain p-values. We then adjusted the resulting p-values using the Benjamini-Hochberg correction (*Benjamini and Hochberg, 1995*).

## Simulation of independent evolution of sites

To model independent evolution of sites, we used genome-wide forward simulator MimicrEE2 (*Vlachos and Kofler, 2018*). Under independent mode of evolution, MimicrEE2 multiplies the fitness changes caused by individual mutations to get the fitness of a genome. The initial population consisted of 50,000 identical haploid genotypes with 100 biallelic (*a* or *A*) sites. At the start of simulation, 20 sites were under positive selection, since the initial allele was deleterious: its fitness was equal to 0.9945, while fitness of the other variant was equal to 1. Another 20 sites evolved under negative selection, since the initial allele was beneficial. The remaining 60 sites evolved neutrally, with the two possible variants at each site having fitness of 1. We simulated evolution of the population for 5000 generations, with mutation rate 5e-4 mutations per site per generation. Each 250 generations, 50 genotypes were sampled from the population, resulting in 1000 sequences. These were further used to reconstruct the phylogeny and measure the signal of epistasis. For each of the four detection models, the minimal FDR threshold was obtained, such that if the desired level of FDR would be below the threshold, no false concordantly evolving pairs of sites would be predicted.

## Simulation of positively and negatively epistatically evolved site pairs

MimicrEE2 allows modeling of epistatic interaction between a pair of sites, assigning fitness values to all possible combinations of binary variants at these sites (*aA*, *aA*, *Aa* and *AA*). The fitness of a genome in this case is the product of fitness values of individual changes and fitnesses of variant combinations for specified pairs. The initial population consisted of identical genotypes with lowercase alleles at each site, with a total of 100 sites: 20 sites (or 10 pairs of sites) in positive epistasis, 20 sites in negative epistasis and 60 neutrally evolving sites with no epistatic interactions. At neutrally evolving sites, all variants had fitness equal to 1. To model positive epistasis between a pair of sites, we assigned fitness 1 to variant combinations *aa* and *AA* and fitness 0.9945 to *aA* and *Aa*, so that the first mutation at one of the sites was deleterious, and the consequent mutation at the second site restored the initial fitness. To model negative epistasis between a pair of sites, we assigned fitness 1 to variant combinations *aa*, *aA* and *Aa*, and fitness 0.8 to *AA*, so that the first mutation at one of the sites was neutral, and the consequent mutation at the second site was deleterious. Again, we simulated evolution of a haploid population of size 50,000 for 5000 generations, with mutation rate 5e-4 mutations per site per generation, sampled 50 genotypes each 250 generations, and used the resulting 1000 sequences to reconstruct the phylogeny and measure the signal of epistasis.

## Comparing the signal strength of coordinated evolution across site subsets

To compare the strength of concordant evolution for pairs of sites belonging to a specified site subset with other pairs of sites, we designed a test comparing the average ranks of pairs of sites in these two subsets of pairs. First, we ordered all site pairs from high to low strength of concordant evolution of their sites according to ascending upper p-values. Then, we calculated the mean ranks of site pairs in each of the two subsets: in a subset of pairs for which both sites belong to the specified subset of sites, and in the complementary subset of pairs. A direct comparison of ranks in these two subsets of pairs may be misleading, because a test for coordinated evolution of sites may assign better p-values to pairs of sites having some properties, for example those with higher evolutionary rates, which could be overrepresented in a specified subset of sites. Therefore, we need to compare the mean ranks for the bipartition of site pairs for the actual data with the distributions of mean ranks for the same bipartition for data simulating independent evolution of sites. As simulated data, we used a set of 400 permutations of mutations on the tree used for the FDR estimation. Calculating the ranks of pairs, we then assigned the average values of ranks for site pairs having the same values of the ordering statistic. As the test statistic, we used the difference of the mean ranks of pairs in two parts of the

bipartition. To calculate the p-value of the test, the test statistic for the actual data was compared with the test statistic obtained for the simulated data.

We applied this procedure to separately tested subsets of lineage-defining sites of VOCs that circulated before May 2021: Alpha (B.1.1.7+Q.*), Beta (B.1.351.*), Delta (B.1.617.2+AY.*) and Gamma (P.1.*). The list of lineage-defining mutations in the S-gene was compiled according to *Hodcroft, 2021* (accessed 23.07.2022). We considered only missense lineage-defining mutations and excluded from the analysis the site S:614, because the mutation D614G was fixed in all considered VOCs.

To test whether the enrichment of pairs of lineage-defining sites for Alpha, Beta, Delta, and Gamma among concordantly evolving site pairs is due to mutations that occurred at origins of these VOCs and/or reversions of these mutations within the VOC clades, for each VOC, we obtained a pruned tree which did not carry the sequences of this VOC, as well as the sequences descendant from the VOC clade but not annotated as this VOC. For this, we removed from the phylogeny the isolates that were assigned by PANGOLIN (*O'Toole et al., 2021*) to the corresponding VOC in their metadata as well as the isolates of other lineages descendant to the ancestral node of the VOC that carried on its branch the earliest lineage-defining mutation. This procedure is conservative, in that it could exclude some of the non-VOC samples that carried a subset of VOC lineage-defining mutations. For each pruned tree, we then applied the procedure of finding concordantly evolving pairs of sites, and then the procedure of comparing the strength of concordant evolution for pairs of lineage-defining sites and the complementary subset of pairs.

## Comparing the sets of concordantly evolving pairs and DCA high scoring pairs

Alignments of protein sequences and DCA scores for pairs of alignment sites for the following PFAM domains of Spike bCoV_S1_N (PF16451.6), bCoV_S1_RBD (PF09408.11), CoV_S1_C (PF19209.1), bCoV_S2 (PF01601.17), CoV_S2_C (PF19214.1) were kindly provided by Dr. Rodriguez-Rivas upon our request. For each alignment, we identified the sequence most similar to the SARS-CoV-2 Spike protein encoded in the genome EPI_ISL_404895 using BLASTP []. The lengths of domains mapped on the SARS-CoV-2 Spike protein were 305 for bCoV_S1_N, 178 for bCoV_S1_RBD, 57 for CoV_S1_C, 519 for bCoV_S2 and 40 for CoV_S2_C. The total number of site pairs for each domain equals $N_D(N_D-1)/2$, where $N_D$ is the number of sites in the domain. For each domain, we considered only those concordantly evolving pairs of sites for which both sites were located within the mapped domains; there were 15 such pairs in bCoV_S1_N, fourteen in bCoV_S1_RBD, zero in CoV_S1_C, four in bCoV_S2, and one in CoV_S2_C. We estimated the chance to find the observed numbers of concordantly evolving pairs of sites among the $N_D$ pairs having the highest DCA scores using Fisher's exact test.

## Acknowledgements

The DCA scores for pairs of sites located within SARS-CoV-2 domains inferred in the study (*Rodriguez-Rivas et al., 2022*) were kindly provided by Dr. Juan Rodriguez-Rivas upon our request. Bioinformatic data analysis was funded by the Russian Science Foundation [grant number 21-74-20160 to GAB]. Simulation analysis was supported in the framework of the HSE University Basic Research Program [to ADN].

## Additional information

### Funding

| Funder | Grant reference number | Author |
| --- | --- | --- |
| Higher School of Economics University Basic Research Program | | Alexey Dmitrievich Neverov |
| Russian Science Foundation | 21-74-20160 | Georgii Bazykin |

| Funder | Grant reference number | Author |
|---|---|---|

The funders had no role in study design, data collection and interpretation, or the decision to submit the work for publication.

## Author contributions

Alexey Dmitrievich Neverov, Software, Formal analysis, Investigation, Methodology, Writing - original draft, Writing - review and editing; Gennady Fedonin, Software, Formal analysis, Methodology; Anfisa Popova, Software, Visualization, Writing - original draft, Writing - review and editing; Daria Bykova, Formal analysis; Georgii Bazykin, Conceptualization, Supervision, Methodology, Writing - original draft, Writing - review and editing

## Author ORCIDs

Alexey Dmitrievich Neverov http://orcid.org/0000-0002-3594-1682

## Decision letter and Author response

Decision letter https://doi.org/10.7554/eLife.82516.sa1
Author response https://doi.org/10.7554/eLife.82516.sa2

## Additional files

### Supplementary files
• MDAR checklist

### Data availability

The codes and data files required to reproduce the analysis are available at https://github.com/gFedonin/EpiStat, (copy archived at swh:1:rev:6c528ec4991854467e98684aa471a9b8f095b875). The data for reproducing analyses from the paper are in the archive file - "sars-2.epistat.data.tgz". All genome sequences and associated metadata in the dataset EPI_SET_20220729hq (Appendix 1 - table 22) are published in GISAID's EpiCoV database. To view the contributors of each individual sequence with details such as accession number, virus name, collection date, originating lab and submitting lab and the list of authors, visit https://doi.org/10.55876/gis8.220729hq.

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

# Appendix 1

**Appendix 1—table 1.** Levels of spurious signal of concordant evolution, inferred in the simulated dataset with no epistasis by four variants of the method.

Number of pairs, detected under p-value threshold $10^{-4}$, and estimated FDR for this threshold are shown.

| Model | | #FP | min p-value | Maximal FDR threshold for #FP = 0 |
|---|---|---|---|---|
| consider alleles | shuffle mutations in fgr. only | 2 | <1e-4 | 0,25 |
| | shuffle mutations both in bgr. and fgr. | 3 | <1e-4 | 0,15 |
| ignore alleles | shuffle mutations in fgr. only | 4 | <1e-4 | 0,11 |
| | shuffle mutations both in bgr. and fgr. | 24 | <1e-4 | 0,02 |

**Appendix 1—table 2.** Levels of spurious signal of discordant evolution, inferred in the simulated dataset with no epistasis by four variants of the method.

Number of pairs, detected under p-value threshold $10^{-4}$, and maximal threshold on estimated FDR that allows to avoid false discoveries are shown.

| Model | | #FP | min p-value | Maximal FDR threshold for #FP = 0 |
|---|---|---|---|---|
| consider alleles | shuffle mutations in fgr. only | 2 | <1e-4 | 0,2 |
| | shuffle mutations both in bgr. and fgr. | 19 | <1e-4 | 0,026 |
| ignore alleles | shuffle mutations in fgr. only | 4 | <1e-4 | 0,125 |
| | shuffle mutations both in bgr. and fgr. | 16 | <1e-4 | 0,031 |

**Appendix 1—table 3.** Numbers of truly and falsely predicted concordantly evolving pairs of sites for the simulated data with positively and negatively epistatically interacting sites.

The evolution of the population of genotypes with twenty independently evolving pairs of epistatically interacting sites was simulated by MimicrEE2. Ten pairs of sites were evolving under recurrent positive epistasis and other ten pairs were evolving under magnitude negative epistasis. The four different detection models were applied and for each model the number of true (#TP) and false (#FP) predictions are shown for estimated FDR≤10%.

| Model | | #FP | #TP |
|---|---|---|---|
| consider alleles | shuffle mutations in fgr. only | 5 | 10 |
| | shuffle mutations both in bgr. and fgr. | 55 | 6 |
| ignore alleles | shuffle mutations in fgr. only | 22 | 8 |
| | shuffle mutations both in bgr. and fgr. | 250 | 7 |

**Appendix 1—table 4.** Predicted concordantly evolving pairs of sites for the simulated data with positively and negatively epistatically interacting sites.

The characteristics of predicted site pairs for the estimated FDR≤10% are shown: positions of sites on the primary sequence of S-protein (site1 and site2), the upper p-values (p-value), values of the epistatic statistic (epistat), numbers of consecutive pairs of mutations (#consec. pairs of mutations), numbers of mutations in consecutive pairs of sites (#mut. in consec. pairs in site1 and site2), total numbers of mutations on the internal tree branches in sites (#mut. in site1 and site2), the expected value of epistatic statistics (exp. epistat) and the corresponding standard error (SE). The true predictions are highlighted in bold.

| site1 | site2 | p-value | epistat | #consec. pairs of mut. | #mut. in consec. pairs in site1 | #mut. in consec. pairs in site2 | #mut. in site1 | #mut. in site2 | exp. epistat | SE |
|---|---|---|---|---|---|---|---|---|---|---|
| 1 | 2 | <1e-4 | 12,36 | 66,5 | 52 | 65 | 2 | 96 | 3,97 | 0,87 |
| 3 | 4 | <1e-4 | 17,79 | 94 | 61 | 104 | 4 | 118 | 6,55 | 1,23 |
| 5 | 6 | <1e-4 | 13,87 | 54 | 48 | 54 | 6 | 105 | 3,27 | 0,83 |
| 5 | 43 | <1e-4 | 21,43 | 108,5 | 82 | 100 | 43 | 105 | 12,6 | 1,89 |
| 7 | 8 | <1e-4 | 11,36 | 50 | 42 | 50 | 8 | 109 | 3,22 | 0,78 |
| 9 | 10 | <1e-4 | 17,09 | 90 | 64 | 88 | 10 | 117 | 6,62 | 1,19 |
| 11 | 12 | <1e-4 | 13,46 | 53,5 | 48 | 51 | 12 | 108 | 2,54 | 0,69 |
| 13 | 14 | <1e-4 | 11,63 | 52,5 | 43 | 56 | 14 | 106 | 2,85 | 0,72 |
| 15 | 16 | <1e-4 | 11,84 | 69 | 47 | 70 | 16 | 116 | 5,88 | 1,08 |
| 19 | 20 | <1e-4 | 9,06 | 47 | 39 | 47 | 18 | 109 | 4,29 | 0,88 |
| 48 | 68 | <1e-4 | 29,11 | 171,5 | 110 | 169 | 20 | 95 | 2,6 | 0,69 |
| 17 | 18 | 2e-4 | 8,23 | 66 | 38 | 65 | 31 | 95 | 6,61 | 1,23 |
| 19 | 31 | 2e-4 | 11,58 | 78,5 | 56 | 78 | 67 | 176 | 20,72 | 2,09 |
| 50 | 94 | 2e-4 | 24,34 | 162,5 | 100 | 151 | 68 | 183 | 17,55 | 1,9 |
| 78 | 94 | 2e-4 | 28,50 | 172 | 110 | 169 | 94 | 178 | 20,5 | 2,07 |

**Appendix 1—table 5.** Numbers of truly and falsely predicted discordantly evolving pairs of sites for the simulated data with positively and negatively epistatically interacting sites.
See legend for *Appendix 1—table 3*.

| Model | | #FP | #TP |
|---|---|---|---|
| | shuffle mutations in fgr. only | 4 | 7 |
| consider alleles | shuffle mutations both in bgr. and fgr. | 62 | 5 |
| | shuffle mutations in fgr. only | 4 | 9 |
| ignore alleles | shuffle mutations both in bgr. and fgr. | 7 | 5 |

**Appendix 1—table 6.** Predicted discordantly evolving pairs for the simulated data with positively and negatively epistatically interacting sites.
See legend for *Appendix 1—table 4*. The 'p-value' column contains the lower p-values.

| site1 | site2 | p-value | epistat | #consec. pairs of mut. | #mut. in consec. pairs in site1 | #mut. in consec. pairs in site2 | #mut. in site1 | #mut. in site2 | exp. epistat | SE |
|---|---|---|---|---|---|---|---|---|---|---|
| 16 | 50 | <1e-4 | 5,72 | 85,5 | 47 | 83 | 108 | 178 | 10,51 | 1,48 |
| 17 | 96 | 1e-4 | 7,44 | 93,5 | 54 | 90 | 109 | 192 | 12,92 | 1,70 |
| 23 | 24 | <1e-4 | 8,27 | 70,5 | 51 | 68 | 137 | 129 | 13,70 | 1,65 |
| 25 | 26 | <1e-4 | 6,13 | 73,5 | 37 | 74 | 139 | 121 | 10,84 | 1,46 |
| 26 | 95 | 1e-4 | 6,94 | 91,5 | 52 | 94 | 121 | 179 | 12,18 | 1,60 |
| 27 | 28 | <1e-4 | 5,90 | 76 | 42 | 73 | 168 | 149 | 13,90 | 1,82 |
| 33 | 34 | 1e-4 | 4,70 | 48,5 | 37 | 45 | 117 | 136 | 8,55 | 1,30 |
| 35 | 36 | <1e-4 | 5,47 | 59,5 | 39 | 55 | 150 | 127 | 10,81 | 1,51 |
| 37 | 38 | <1e-4 | 4,10 | 44,5 | 32 | 41 | 113 | 146 | 8,93 | 1,39 |

*Appendix 1—table 6 Continued on next page*

*Appendix 1—table 6 Continued*

| site1 | site2 | p-value | epistat | #consec. pairs of mut. | #mut. in consec. pairs in site1 | #mut. in consec. pairs in site2 | #mut. in site1 | #mut. in site2 | exp. epistat | SE |
|---|---|---|---|---|---|---|---|---|---|---|
| 39 | 40 | <1e-4 | 4,97 | 61,5 | 37 | 59 | 115 | 142 | 11,11 | 1,47 |
| 39 | 47 | <1e-4 | 7,27 | 86,5 | 54 | 84 | 115 | 166 | 13,09 | 1,66 |

**Appendix 1—table 7.** Predicted concordantly evolving pairs for the ML phylogeny of S-gene reconstructed by IQ-TREE.

The characteristics of predicted site pairs for the estimated FDR≤10% are shown: the positions of sites on the primary sequence of S-protein (site1 and site2), the upper p-values (p-value), values of the epistatic statistic (epistat), numbers of consecutive pairs of mutations (#consec. pairs of mutations), numbers of mutations in consecutive pairs of sites (#mut. in consec. pairs in site1 and site2) and total numbers of mutations on the internal tree branches in sites (#mut. in site1 and site2), FDR value corresponding to the p-value of the site pair obtained for the alternative phylogeny reconstructed by USHER and distances between sites on the protein structure 7JJJ (pdb distance).

| site1 | site2 | p-value | epistat | #consec. pairs of mut. | #mut. in consec. pairs in site1 | #mut. in consec. pairs in site2 | #mut. in site1 | #mut. in site2 | FDR for the USHER tree | pdb distance |
|---|---|---|---|---|---|---|---|---|---|---|
| 13 | 152 | <2e-5 | 2,864 | 5 | 5 | 4 | 5 | 16 | 0,03 | |
| 18 | 20 | 0,00008 | 1,661 | 3,5 | 3 | 3 | 28 | 8 | 1,19 | 3,62 |
| 20 | 26 | <2e-5 | 1,985 | 4 | 3 | 3 | 8 | 12 | 0,36 | 15,19 |
| 20 | 417 | 0,00022 | 1,348 | 4 | 3 | 2 | 8 | 7 | 0,05 | 47,83 |
| 26 | 190 | 0,00018 | 1,029 | 4 | 4 | 3 | 12 | 3 | 0,06 | 18,40 |
| 63 | 64 | 0,00002 | 0,726 | 1 | 1 | 1 | 2 | 4 | 1,00 | 1,31 |
| 63 | 67 | 0,00008 | 0,726 | 1 | 1 | 1 | 2 | 5 | 1,00 | 9,38 |
| 63 | 69 | 0,00012 | 0,726 | 1 | 1 | 1 | 2 | 5 | 1,00 | 11,07 |
| 63 | 213 | 0,00004 | 0,681 | 1 | 1 | 1 | 2 | 3 | 0,01 | 13,51 |
| 64 | 67 | 0,00012 | 0,726 | 1 | 1 | 1 | 4 | 5 | 1,00 | 5,51 |
| 64 | 69 | 0,00014 | 0,726 | 1 | 1 | 1 | 4 | 5 | 1,00 | 7,05 |
| 67 | 69 | <2e-5 | 1,101 | 2 | 2 | 2 | 5 | 5 | 0,55 | 3,59 |
| 69 | 70 | <2e-5 | 1,125 | 2 | 2 | 2 | 5 | 4 | 0,01 | 1,30 |
| 70 | 144 | <2e-5 | 1,301 | 3 | 3 | 3 | 4 | 5 | 0,01 | 14,03 |
| 76 | 490 | 0,00026 | 0,220 | 1 | 1 | 1 | 3 | 3 | 0,05 | 45,18 |
| 154 | 1071 | <2e-5 | 1,767 | 4 | 2 | 3 | 5 | 5 | 0,25 | 95,13 |
| 155 | 157 | <2e-5 | 1,113 | 2 | 2 | 2 | 3 | 11 | 0,53 | 3,74 |
| 189 | 356 | 0,00006 | 0,544 | 2 | 1 | 2 | 3 | 2 | 0,32 | 31,63 |
| 189 | 360 | 0,00016 | 0,544 | 2 | 1 | 2 | 3 | 3 | 0,07 | 29,89 |
| 190 | 417 | 0,0001 | 1,272 | 3 | 2 | 2 | 3 | 7 | 0,01 | 39,93 |
| 213 | 261 | 0,00008 | 0,562 | 1 | 1 | 1 | 3 | 2 | 0,37 | 11,10 |
| 259 | 261 | <2e-5 | 1,350 | 1,5 | 2 | 1 | 4 | 2 | 0,05 | 3,81 |
| 262 | 272 | <2e-5 | 1,183 | 4 | 4 | 1 | 9 | 2 | 0,94 | 31,42 |
| 356 | 357 | 0,00006 | 0,448 | 1 | 1 | 1 | 2 | 2 | 0,69 | 1,33 |
| 356 | 360 | <2e-5 | 1,283 | 2,5 | 2 | 3 | 2 | 3 | 0,05 | 10,12 |

*Appendix 1—table 7 Continued on next page*

*Appendix 1—table 7 Continued*

| site1 | site2 | p-value | epistat | #consec. pairs of mut. | #mut. in consec. pairs in site1 | #mut. in consec. pairs in site2 | #mut. in site1 | #mut. in site2 | FDR for the USHER tree | pdb distance |
|---|---|---|---|---|---|---|---|---|---|---|
| 359 | 360 | <2e-5 | 0,976 | 1,5 | 1 | 2 | 2 | 3 | 0,01 | 1,34 |
| 439 | 441 | <2e-5 | 2,097 | 4 | 3 | 4 | 9 | 8 | 0,01 | 3,50 |
| 440 | 441 | 0,00014 | 1,505 | 3 | 3 | 3 | 12 | 8 | 0,01 | 1,33 |
| 440 | 442 | <2e-5 | 2,476 | 3 | 2 | 3 | 12 | 6 | 0,01 | 3,92 |
| 440 | 443 | 0,00002 | 1,284 | 4 | 3 | 2 | 12 | 2 | 0,25 | 3,86 |
| 440 | 444 | <2e-5 | 2,515 | 3,5 | 2 | 5 | 12 | 7 | 0,01 | 5,60 |
| 441 | 442 | <2e-5 | 2,244 | 4 | 4 | 4 | 8 | 6 | 0,01 | 1,33 |
| 441 | 443 | 0,00002 | 1,281 | 5 | 4 | 2 | 8 | 2 | 0,02 | 3,23 |
| 441 | 444 | <2e-5 | 3,492 | 5,5 | 4 | 4 | 8 | 7 | 0,01 | 2,60 |
| 442 | 443 | 0,00006 | 1,301 | 4 | 4 | 2 | 6 | 2 | 0,05 | 1,32 |
| 442 | 444 | <2e-5 | 3,013 | 6 | 4 | 4 | 6 | 7 | 0,01 | 4,05 |
| 443 | 444 | 0,00008 | 1,043 | 2 | 2 | 2 | 2 | 7 | 0,03 | 1,32 |
| 484 | 655 | 0,00004 | 1,716 | 8 | 6 | 5 | 34 | 12 | 0,89 | 73,58 |
| 501 | 1118 | 0,00014 | 2,737 | 16 | 13 | 12 | 40 | 22 | 0,09 | 131,71 |
| 681 | 716 | <2e-5 | 3,905 | 16,5 | 12 | 16 | 59 | 21 | 0,02 | |
| 716 | 982 | 0,00002 | 2,001 | 15 | 9 | 11 | 21 | 15 | 0,01 | 81,66 |
| 716 | 1118 | 0,00004 | 2,382 | 13 | 8 | 12 | 21 | 22 | 0,01 | 22,29 |
| 859 | 950 | <2e-5 | 2,219 | 5,5 | 4 | 5 | 11 | 9 | 0,01 | 15,17 |
| 982 | 1118 | 0,00014 | 1,637 | 15 | 11 | 8 | 15 | 22 | 0,01 | 94,63 |
| 1258 | 1259 | 0,00004 | 0,775 | 3 | 1 | 3 | 5 | 3 | NA | |

**Appendix 1—table 8.** Predicted concordantly evolving pairs for the MP phylogeny of S-gene reconstructed by USHER.

See legend for *Appendix 1—table 7*. The 'p-value' column contains the lower p-values.

| site1 | site2 | p-value | epistat | #consec. pairs of mut. | #mut. in consec. pairs in site1 | #mut. in consec. pairs in site2 | #mut. in site1 | #mut. in site2 | IQ-TREE | pdb distance |
|---|---|---|---|---|---|---|---|---|---|---|
| 12 | 346 | 0,00006 | 0,934 | 2 | 2 | 1 | 9 | 4 | 0 | - |
| 12 | 899 | 0,00004 | 0,724 | 2 | 2 | 1 | 9 | 3 | 0 | - |
| 13 | 152 | 0,00004 | 1,407 | 5 | 4 | 3 | 4 | 13 | 1 | - |
| 20 | 190 | 0,00022 | 0,885 | 3 | 2 | 3 | 8 | 4 | 0 | 22,59 |
| 20 | 417 | 0,00018 | 1,107 | 2 | 2 | 1 | 8 | 8 | 1 | 47,83 |
| 26 | 190 | 0,00026 | 0,913 | 4 | 3 | 4 | 12 | 4 | 1 | 18,40 |
| 26 | 655 | 0,00042 | 0,968 | 6,5 | 3 | 6 | 12 | 14 | 0 | 37,53 |
| 54 | 690 | 0,00044 | 0,714 | 1 | 1 | 1 | 11 | 3 | 0 | 33,65 |
| 62 | 251 | <2e-5 | 0,748 | 1 | 1 | 1 | 2 | 3 | 0 | 32,01 |
| 67 | 96 | <2e-5 | 0,599 | 1 | 1 | 1 | 3 | 7 | 0 | 7,12 |
| 69 | 70 | <2e-5 | 1,496 | 3 | 3 | 3 | 5 | 5 | 1 | 1,30 |

*Appendix 1—table 8 Continued on next page*

*Appendix 1—table 8 Continued*

| site1 | site2 | p-value | epistat | #consec. pairs of mut. | #mut. in consec. pairs in site1 | #mut. in consec. pairs in site2 | #mut. in site1 | #mut. in site2 | IQ-TREE | pdb distance |
|---|---|---|---|---|---|---|---|---|---|---|
| 69 | 144 | 0,00002 | 0,984 | 2 | 2 | 2 | 5 | 5 | 0 | 12,57 |
| 70 | 144 | <2e-5 | 1,088 | 3 | 3 | 3 | 5 | 5 | 1 | 14,03 |
| 76 | 490 | 0,0001 | 0,275 | 1 | 1 | 1 | 2 | 3 | 1 | 45,18 |
| 80 | 215 | <2e-5 | 1,687 | 6,5 | 3 | 6 | 21 | 14 | 0 | 12,51 |
| 80 | 950 | 0,00014 | 1,091 | 3 | 2 | 2 | 21 | 7 | 0 | 56,59 |
| 152 | 252 | 0,00016 | 0,623 | 2 | 2 | 1 | 13 | 4 | 0 | 15,26 |
| 189 | 360 | 0,0003 | 0,534 | 1 | 1 | 1 | 2 | 2 | 1 | 29,89 |
| 189 | 772 | 0,00024 | 0,440 | 1 | 1 | 1 | 2 | 2 | 0 | 44,17 |
| 190 | 417 | <2e-5 | 1,134 | 2,5 | 2 | 2 | 4 | 8 | 1 | 39,93 |
| 215 | 1167 | 0,00042 | 0,853 | 2 | 2 | 2 | 14 | 4 | 0 | |
| 255 | 256 | 0,00022 | 0,758 | 2 | 2 | 2 | 8 | 7 | 0 | 1,32 |
| 255 | 260 | 0,00036 | 0,811 | 3 | 3 | 2 | 8 | 3 | 0 | 9,61 |
| 256 | 258 | 0,00012 | 0,660 | 1 | 1 | 1 | 7 | 2 | 0 | 2,81 |
| 256 | 260 | 0,00014 | 1,056 | 2 | 2 | 2 | 7 | 3 | 0 | 8,18 |
| 259 | 260 | <2e-5 | 1,536 | 2 | 2 | 2 | 3 | 3 | 0 | 1,31 |
| 259 | 261 | 0,00014 | 0,909 | 1 | 1 | 1 | 3 | 2 | 1 | 3,81 |
| 357 | 360 | <2e-5 | 1,025 | 1,5 | 1 | 2 | 2 | 2 | 0 | 7,59 |
| 359 | 360 | <2e-5 | 1,268 | 2 | 1 | 2 | 2 | 2 | 1 | 1,34 |
| 360 | 772 | 0,00018 | 0,534 | 1 | 1 | 1 | 2 | 2 | 0 | 58,61 |
| 439 | 440 | <2e-5 | 2,038 | 3,5 | 2 | 4 | 10 | 12 | 0 | 1,31 |
| 439 | 441 | <2e-5 | 2,795 | 5 | 3 | 4 | 10 | 5 | 1 | 3,50 |
| 439 | 444 | 0,00006 | 1,606 | 4,5 | 3 | 4 | 10 | 7 | 0 | 6,21 |
| 440 | 441 | <2e-5 | 2,822 | 4,5 | 4 | 2 | 12 | 5 | 1 | 1,33 |
| 440 | 442 | <2e-5 | 2,092 | 3 | 3 | 2 | 12 | 3 | 1 | 3,92 |
| 440 | 444 | <2e-5 | 2,947 | 4,5 | 4 | 3 | 12 | 7 | 1 | 5,60 |
| 441 | 442 | <2e-5 | 2,292 | 3 | 2 | 2 | 5 | 3 | 1 | 1,33 |
| 441 | 443 | 0,00002 | 1,383 | 2 | 2 | 2 | 5 | 2 | 1 | 3,23 |
| 441 | 444 | <2e-5 | 3,829 | 6,5 | 4 | 4 | 5 | 7 | 1 | 2,60 |
| 441 | 445 | 0,00018 | 1,086 | 2 | 2 | 1 | 5 | 6 | 0 | 8,51 |
| 442 | 443 | 0,0001 | 1,132 | 2 | 2 | 1 | 3 | 2 | 1 | 1,32 |
| 442 | 444 | 0 | 2,387 | 4 | 3 | 3 | 3 | 7 | 1 | 4,05 |
| 443 | 444 | 0,00006 | 1,031 | 4 | 1 | 4 | 2 | 7 | 1 | 1,32 |
| 444 | 445 | 0,0002 | 1,270 | 4 | 5 | 3 | 7 | 6 | 0 | 1,31 |
| 501 | 570 | 0 | 2,734 | 16 | 13 | 10 | 46 | 19 | 0 | 53,69 |
| 501 | 1118 | 0,00048 | 2,028 | 18 | 14 | 13 | 46 | 19 | 1 | 131,71 |
| 570 | 681 | 0,00018 | 2,538 | 17,5 | 15 | 14 | 19 | 62 | 0 | |

*Appendix 1—table 8 Continued on next page*

*Appendix 1—table 8 Continued*

| site1 | site2 | p-value | epistat | #consec. pairs of mut. | #mut. in consec. pairs in site1 | #mut. in consec. pairs in site2 | #mut. in site1 | #mut. in site2 | IQ-TREE | pdb distance |
|---|---|---|---|---|---|---|---|---|---|---|
| 570 | 1118 | 0,00002 | 1,881 | 13,5 | 9 | 10 | 19 | 19 | 0 | 79,04 |
| 572 | 1181 | 0 | 0,533 | 1 | 1 | 1 | 3 | 2 | 0 | |
| 583 | 1237 | 0,00014 | 0,242 | 1 | 1 | 1 | 3 | 3 | 1 | |
| 681 | 716 | 0,00002 | 3,017 | 20 | 20 | 16 | 62 | 21 | 1 | |
| 681 | 1118 | 0,00032 | 2,334 | 18 | 14 | 12 | 62 | 19 | 0 | |
| 716 | 982 | 0 | 4,623 | 19,5 | 13 | 16 | 21 | 18 | 1 | 81,66 |
| 716 | 1118 | 0 | 2,872 | 17,5 | 11 | 13 | 21 | 19 | 1 | 22,29 |
| 859 | 950 | 0 | 1,596 | 4 | 3 | 4 | 9 | 7 | 1 | 15,17 |
| 982 | 1118 | 0 | 1,831 | 17,5 | 10 | 10 | 18 | 19 | 1 | 94,63 |
| 1027 | 1176 | 0,00016 | 1,206 | 5,5 | 5 | 4 | 12 | 9 | 0 | |

**Appendix 1—table 9.** Predicted discordantly evolving pairs for the MP phylogeny of S-gene reconstructed by USHER.

The characteristics of predicted site pairs for the estimated FDR≤10% are shown: the positions of sites on the primary sequence of S-protein (site1 and site2), the upper p-values (p-value), values of the epistatic statistic (epistat), numbers of consecutive pairs of mutations (#consec. pairs of mutations), numbers of mutations in consecutive pairs of sites (#mut. in consec. pairs in site1 and site2) and total numbers of mutations on the internal tree branches in sites (#mut. in site1 and site2), indicator variable that marks whether a site pairs is also within the set of predictions for the ML phylogeny reconstructed by IQ-TREE for the same FDR threshold (IQ-TREE tree) and distances between sites on the protein structure (pdb distance).

| site1 | site2 | p-value | epistat | #consec. pairs of mut. | #mut. in consec. pairs in site1 | #mut. in consec. pairs in site2 | #mut. in site1 | #mut. in site2 | IQ-TREE | pdb distance |
|---|---|---|---|---|---|---|---|---|---|---|
| 18 | 681 | 0,00344 | 0,102 | 5,5 | 5 | 4 | 22 | 62 | 0 | |
| 222 | 501 | 0,00314 | 0,019 | 1 | 1 | 1 | 12 | 46 | 1 | 55,44 |
| 439 | 501 | 0,00286 | 0 | 0 | 0 | 0 | 10 | 46 | 0 | 5,16 |
| 440 | 501 | 0,0019 | 0 | 0 | 0 | 0 | 12 | 46 | 0 | 9,47 |
| 440 | 681 | 0,00274 | 0,008 | 1 | 1 | 1 | 12 | 62 | 1 | |
| 484 | 982 | 0,0021 | 0,020 | 3 | 3 | 3 | 43 | 18 | 0 | 35,42 |
| 501 | 675 | 0,00136 | 0,025 | 1 | 1 | 1 | 46 | 22 | 1 | 84,98 |
| 501 | 677 | 0,0001 | 0,023 | 2 | 2 | 2 | 46 | 33 | 1 | 88,2 |
| 570 | 677 | 0,00226 | 0,011 | 1 | 1 | 1 | 19 | 33 | 0 | 44,77 |
| 614 | 653 | 0,00022 | 0,037 | 4 | 1 | 4 | 10 | 5 | 1 | 15,47 |
| 675 | 716 | 0,00568 | 0 | 0 | 0 | 0 | 22 | 21 | 0 | 38,15 |
| 675 | 1118 | 0,00524 | 0 | 0 | 0 | 0 | 22 | 19 | 0 | 58,84 |
| 677 | 681 | 0,00264 | 0,098 | 3 | 3 | 3 | 33 | 62 | 0 | |
| 677 | 716 | 0,00102 | 0,011 | 1 | 1 | 1 | 33 | 21 | 0 | 42,93 |
| 677 | 982 | 0,00152 | 0,004 | 1 | 1 | 1 | 33 | 18 | 0 | 54,37 |
| 677 | 1118 | 0,00066 | 0,002 | 1 | 1 | 1 | 33 | 19 | 0 | 64,36 |

**Appendix 1—table 10.** Coordinated episodic selection in concordantly evolving pairs predicted for the phylogeny reconstructed by IQ-TREE.

Z-scores < 0 and upper p-values after Benjamini-Hochberg adjustment < 0.05 indicate clustering of nonconsecutive mutations; z-scores > 0 and lower p-values after Benjamini-Hochberg adjustment < 0.05 indicate that nonconsecutive mutations tend to avoid each other. For site pairs with z-scores > 0, concordant evolution cannot be explained without positive epistatic interaction between the sites.

| site 1 | site 2 | #nonseq. pairs | zscore | lower pvalue | upper pvalue | lower pvalue adj. | upper pvalue adj. |
|---|---|---|---|---|---|---|---|
| 13 | 152 | 2569 | –0,724 | 0,7675 | 0,2325 | 1 | 0,5813 |
| 18 | 20 | 8610,5 | 1,148 | 0,1225 | 0,8775 | 0,3675 | 1 |
| 20 | 26 | 5719 | 2,444 | 0,0175 | 0,9825 | 0,1125 | 1 |
| 20 | 417 | 2533 | 1,738 | 0,04 | 0,96 | 0,2 | 1 |
| 26 | 190 | 3682 | 2,258 | 0,015 | 0,985 | 0,1125 | 1 |
| 63 | 64 | 124 | 1,736 | 0,06 | 0,94 | 0,2455 | 1 |
| 63 | 67 | 264 | 1,68 | 0,075 | 0,925 | 0,2596 | 1 |
| 63 | 69 | 239 | 2,224 | 0,025 | 0,975 | 0,1406 | 1 |
| 63 | 213 | 94 | 1,407 | 0,0825 | 0,9175 | 0,2652 | 1 |
| 64 | 67 | 1324 | 2,353 | 0,0125 | 0,9875 | 0,1125 | 1 |
| 64 | 69 | 1199 | 2,766 | 0,0025 | 0,9975 | 0,0281 | 1 |
| 67 | 69 | 2542 | 4,104 | 0 | 1 | 0,0281 | 1 |
| 69 | 70 | 2014 | 4,041 | 0 | 1 | 0,0281 | 1 |
| 70 | 144 | 627 | 2,952 | 0,0025 | 0,9975 | 0,0281 | 1 |
| 76 | 490 | 1748 | 0,632 | 0,25 | 0,75 | 0,5357 | 1 |
| 154 | 1071 | 502 | –1,357 | 0,925 | 0,075 | 1 | 0,225 |
| 155 | 157 | 482 | 0,671 | 0,2375 | 0,7625 | 0,5344 | 1 |
| 189 | 356 | 108 | –0,187 | 0,555 | 0,445 | 0,999 | 0,9536 |
| 189 | 360 | 58 | 0,927 | 0,17 | 0,83 | 0,4765 | 1 |
| 190 | 417 | 1631 | 0,099 | 0,445 | 0,555 | 0,8344 | 1 |
| 213 | 261 | 778 | –1,251 | 0,905 | 0,095 | 1 | 0,2672 |
| 259 | 261 | 736,5 | –2,384 | 0,9975 | 0,0025 | 1 | 0,0094 |
| 262 | 272 | 744 | 0,085 | 0,44 | 0,56 | 0,8344 | 1 |
| 356 | 357 | 65 | –0,797 | 0,7525 | 0,2475 | 1 | 0,5862 |
| 356 | 360 | 63,5 | –2,704 | 1 | 0,0025 | 1 | 0,0094 |
| 359 | 360 | 64,5 | –0,39 | 0,6275 | 0,3725 | 1 | 0,8381 |
| 439 | 441 | 1196 | –3,989 | 1 | 0,0025 | 1 | 0,0094 |
| 440 | 441 | 1647 | –3,55 | 1 | 0,0025 | 1 | 0,0094 |
| 440 | 442 | 822 | –3,23 | 1 | 0,0025 | 1 | 0,0094 |
| 440 | 443 | 931 | –2,124 | 0,995 | 0,005 | 1 | 0,0173 |
| 440 | 444 | 1591,5 | –3,511 | 1 | 0,0025 | 1 | 0,0094 |
| 441 | 442 | 446 | –5,063 | 1 | 0,0025 | 1 | 0,0094 |
| 441 | 443 | 505 | –4,26 | 1 | 0,0025 | 1 | 0,0094 |
| 441 | 444 | 864,5 | –5,337 | 1 | 0,0025 | 1 | 0,0094 |
| 442 | 443 | 251 | –3,988 | 1 | 0,0025 | 1 | 0,0094 |

*Appendix 1—table 10 Continued on next page*

*Appendix 1—table 10 Continued*

| site 1 | site 2 | #nonseq. pairs | zscore | lower pvalue | upper pvalue | lower pvalue adj. | upper pvalue adj. |
|---|---|---|---|---|---|---|---|
| 442 | 444 | 429 | −4,727 | 1 | 0,0025 | 1 | 0,0094 |
| 443 | 444 | 491 | −3,972 | 1 | 0,0025 | 1 | 0,0094 |
| 484 | 655 | 18792 | 1,038 | 0,18 | 0,82 | 0,4765 | 1 |
| 501 | 1118 | 16121 | 1,443 | 0,07 | 0,93 | 0,2596 | 1 |
| 681 | 716 | 19375,5 | −0,962 | 0,8325 | 0,1675 | 1 | 0,4434 |
| 716 | 982 | 8267 | 0,58 | 0,3075 | 0,6925 | 0,629 | 1 |
| 716 | 1118 | 9986 | 1,65 | 0,05 | 0,95 | 0,225 | 1 |
| 859 | 950 | 3684,5 | 0,747 | 0,2075 | 0,7925 | 0,5188 | 1 |
| 982 | 1118 | 8103 | 0,757 | 0,2325 | 0,7675 | 0,5344 | 1 |
| 1258 | 1259 | 1053 | −1,435 | 0,9375 | 0,0625 | 1 | 0,2009 |

**Appendix 1—table 11.** Coordinated episodic selection in discordantly evolving pairs predicted for the phylogeny reconstructed by IQ-TREE.

Z-scores < 0 and upper p-values after Benjamini-Hochberg adjustment < 0.05 indicate clustering of nonconsecutive mutations; z-scores > 0 and lower p-values after Benjamini-Hochberg adjustment < 0.05 indicate that nonconsecutive mutations tend to avoid each other. For site pairs with z-scores < 0, discordant evolution cannot be explained without negative epistatic interaction between the sites.

| site 1 | site 2 | #nonseq. pairs | zscore | lower pvalue | upper pvalue | lower pvalue adj. | upper pvalue adj. |
|---|---|---|---|---|---|---|---|
| 69 | 614 | 2875 | 2,27 | 0,015 | 0,985 | 0,06 | 1 |
| 222 | 501 | 17114 | −0,861 | 0,795 | 0,205 | 0,995 | 0,369 |
| 440 | 681 | 10559 | −2,229 | 0,995 | 0,005 | 0,995 | 0,045 |
| 501 | 675 | 18907 | −2,39 | 0,9875 | 0,0125 | 0,995 | 0,0563 |
| 501 | 677 | 30478 | −1,463 | 0,925 | 0,075 | 0,995 | 0,1856 |
| 570 | 614 | 5813 | 1,738 | 0,0425 | 0,9575 | 0,0956 | 1 |
| 614 | 653 | 1856 | 2,944 | 0,0025 | 1 | 0,0225 | 1 |
| 614 | 982 | 4913 | 2,226 | 0,02 | 0,98 | 0,06 | 1 |
| 681 | 1176 | 9599 | −1,326 | 0,9175 | 0,0825 | 0,995 | 0,1856 |

**Appendix 1—table 12.** Coordinated episodic selection in concordantly evolving pairs predicted for the phylogeny reconstructed by USHER.

Z-scores < 0 and upper p-values after Benjamini-Hochberg adjustment < 0.05 indicate clustering of nonconsecutive mutations; z-scores > 0 and lower p-values after Benjamini-Hochberg adjustment < 0.05 indicate that nonconsecutive mutations tend to avoid each other. For site pairs with z-scores > 0, concordant evolution cannot be explained without positive epistatic interaction between the sites.

| site 1 | site 2 | #nonseq. pairs | zscore | lower pvalue | upper pvalue | lower pvalue adj. | upper pvalue adj. |
|---|---|---|---|---|---|---|---|
| 12 | 346 | 1296 | 0,461 | 0,29 | 0,71 | 1 | 0,9263 |
| 12 | 899 | 883 | 0,705 | 0,245 | 0,755 | 1 | 0,9551 |
| 13 | 152 | 3955 | −0,731 | 0,75 | 0,25 | 1 | 0,4597 |
| 20 | 190 | 2357 | −0,372 | 0,6525 | 0,3475 | 1 | 0,5826 |
| 20 | 417 | 3184 | 0,692 | 0,225 | 0,775 | 1 | 0,9551 |
| 26 | 190 | 3876 | 0,993 | 0,1425 | 0,8575 | 0,8835 | 0,9975 |
| 26 | 655 | 10663,5 | 2,486 | 0,01 | 0,99 | 0,1425 | 1 |

*Appendix 1—table 12 Continued on next page*

*Appendix 1—table 12 Continued*

| site 1 | site 2 | #nonseq. pairs | zscore | lower pvalue | upper pvalue | lower pvalue adj. | upper pvalue adj. |
|---|---|---|---|---|---|---|---|
| 54 | 690 | 923 | −1,288 | 0,9175 | 0,0825 | 1 | 0,1959 |
| 62 | 251 | 95 | 0,51 | 0,2875 | 0,7125 | 1 | 0,9263 |
| 67 | 96 | 2027 | 0,462 | 0,285 | 0,715 | 1 | 0,9263 |
| 69 | 70 | 2241 | 2,982 | 0 | 1 | 0,0713 | 1 |
| 69 | 144 | 967 | 2,582 | 0,01 | 0,99 | 0,1425 | 1 |
| 70 | 144 | 833 | 2,303 | 0,02 | 0,98 | 0,1832 | 1 |
| 76 | 490 | 1801 | 0,766 | 0,2125 | 0,7875 | 1 | 0,9551 |
| 80 | 215 | 13261,5 | −2,369 | 0,995 | 0,005 | 1 | 0,0204 |
| 80 | 950 | 6321 | −1,417 | 0,9325 | 0,0675 | 1 | 0,1749 |
| 152 | 252 | 1142 | 0,042 | 0,4875 | 0,5125 | 1 | 0,8063 |
| 189 | 360 | 43 | 0,188 | 0,395 | 0,605 | 1 | 0,8621 |
| 189 | 772 | 109 | −0,85 | 0,785 | 0,215 | 1 | 0,4226 |
| 190 | 417 | 2157,5 | −0,741 | 0,7725 | 0,2275 | 1 | 0,4323 |
| 215 | 1167 | 2031 | −1,602 | 0,9525 | 0,0475 | 1 | 0,1425 |
| 255 | 256 | 2126 | −1,048 | 0,845 | 0,155 | 1 | 0,3398 |
| 255 | 260 | 1285 | −2,55 | 1 | 0,0025 | 1 | 0,0119 |
| 256 | 258 | 645 | −1,441 | 0,9375 | 0,0625 | 1 | 0,1696 |
| 256 | 260 | 872 | −1,368 | 0,9275 | 0,0725 | 1 | 0,1797 |
| 259 | 260 | 435 | −2,326 | 0,9925 | 0,0075 | 1 | 0,0267 |
| 259 | 261 | 778 | −1,861 | 0,99 | 0,01 | 1 | 0,0335 |
| 357 | 360 | 30,5 | −1,517 | 0,95 | 0,05 | 1 | 0,1425 |
| 359 | 360 | 38 | −0,549 | 0,6725 | 0,3275 | 1 | 0,5657 |
| 360 | 772 | 39 | −0,674 | 0,7225 | 0,2775 | 1 | 0,4943 |
| 439 | 440 | 2456,5 | −2,577 | 0,9925 | 0,0075 | 1 | 0,0267 |
| 439 | 441 | 1061 | −2,935 | 1 | 0,0025 | 1 | 0,0119 |
| 439 | 444 | 1143,5 | −3,434 | 1 | 0,0025 | 1 | 0,0119 |
| 440 | 441 | 1555,5 | −2,671 | 0,995 | 0,005 | 1 | 0,0204 |
| 440 | 442 | 897 | −2,686 | 1 | 0,0025 | 1 | 0,0119 |
| 440 | 444 | 1675,5 | −2,976 | 1 | 0,0025 | 1 | 0,0119 |
| 441 | 442 | 387 | −4,065 | 1 | 0,0025 | 1 | 0,0119 |
| 441 | 443 | 440 | −3,437 | 1 | 0,0025 | 1 | 0,0119 |
| 441 | 444 | 721,5 | −4,488 | 1 | 0,0025 | 1 | 0,0119 |
| 441 | 445 | 596 | −2,144 | 0,9825 | 0,0175 | 1 | 0,0554 |
| 442 | 443 | 253 | −3,58 | 1 | 0,0025 | 1 | 0,0119 |
| 442 | 444 | 416 | −4,346 | 1 | 0,0025 | 1 | 0,0119 |
| 443 | 444 | 472 | −4,301 | 1 | 0,0025 | 1 | 0,0119 |
| 444 | 445 | 640 | −2,502 | 0,9975 | 0,0025 | 1 | 0,0119 |
| 501 | 570 | 21489 | 3,943 | 0 | 1 | 0,0713 | 1 |

*Appendix 1—table 12 Continued*

| site 1 | site 2 | #nonseq. pairs | zscore | lower pvalue | upper pvalue | lower pvalue adj. | upper pvalue adj. |
|---|---|---|---|---|---|---|---|
| 501 | 1118 | 21300 | 1,398 | 0,08 | 0,92 | 0,57 | 1 |
| 570 | 681 | 26547,5 | 2,15 | 0,015 | 0,985 | 0,171 | 1 |
| 570 | 1118 | 13096,5 | 1,926 | 0,0225 | 0,9775 | 0,1832 | 1 |
| 572 | 1181 | 511 | –0,81 | 0,785 | 0,215 | 1 | 0,4226 |
| 583 | 1237 | 3077 | –0,998 | 0,8325 | 0,1675 | 1 | 0,3536 |
| 681 | 716 | 27238 | –1,029 | 0,8575 | 0,1425 | 1 | 0,3249 |
| 681 | 1118 | 26316 | 0,042 | 0,475 | 0,525 | 1 | 0,8063 |
| 716 | 982 | 10836,5 | 0,057 | 0,4625 | 0,5375 | 1 | 0,8063 |
| 716 | 1118 | 13434,5 | –0,024 | 0,5175 | 0,4825 | 1 | 0,7858 |
| 859 | 950 | 4892 | 0,429 | 0,3125 | 0,6875 | 1 | 0,9263 |
| 982 | 1118 | 10470,5 | 0,258 | 0,3975 | 0,6025 | 1 | 0,8621 |
| 1027 | 1176 | 4023,5 | 0,98 | 0,155 | 0,845 | 0,8835 | 0,9975 |

**Appendix 1—table 13.** Coordinated episodic selection in discordantly evolving pairs predicted for the phylogeny reconstructed by USHER.

Z-scores < 0 and upper p-values after Benjamini-Hochberg adjustment < 0.05 indicate clustering of nonconsecutive mutations; z-scores > 0 and lower p-values after Benjamini-Hochberg adjustment < 0.05 indicate that nonconsecutive mutations tend to avoid each other. For site pairs with z-scores < 0, discordant evolution cannot be explained without negative epistatic interaction between the sites.

| site 1 | site 2 | #nonseq. pairs | zscore | lower pvalue | upper pvalue | lower pvalue adj. | upper pvalue adj. |
|---|---|---|---|---|---|---|---|
| 18 | 681 | 34644,5 | –1,979 | 0,9725 | 0,0275 | 0,98 | 0,2 |
| 222 | 501 | 20756 | 0,222 | 0,3825 | 0,6175 | 0,68 | 1 |
| 439 | 501 | 7667 | –1,207 | 0,8875 | 0,1125 | 0,98 | 0,45 |
| 440 | 501 | 11220 | 0,422 | 0,325 | 0,675 | 0,665 | 1 |
| 440 | 681 | 13859 | –0,599 | 0,7375 | 0,2625 | 0,98 | 0,6629 |
| 484 | 982 | 21249 | 4,304 | 0,0025 | 1 | 0,0133 | 1 |
| 501 | 675 | 22065 | –1,868 | 0,9625 | 0,0375 | 0,98 | 0,2 |
| 501 | 677 | 36276 | –0,526 | 0,71 | 0,29 | 0,98 | 0,6629 |
| 570 | 677 | 22309 | 4,782 | 0,0025 | 1 | 0,0133 | 1 |
| 614 | 653 | 2273 | 1,739 | 0,05 | 0,95 | 0,16 | 1 |
| 675 | 716 | 13924 | –0,731 | 0,7725 | 0,2275 | 0,98 | 0,6629 |
| 675 | 1118 | 13452 | 0,417 | 0,3325 | 0,6675 | 0,665 | 1 |
| 677 | 681 | 44811 | –2,043 | 0,98 | 0,02 | 0,98 | 0,2 |
| 677 | 716 | 22891 | 0,604 | 0,265 | 0,735 | 0,665 | 1 |
| 677 | 982 | 17847 | 3,543 | 0,0025 | 1 | 0,0133 | 1 |
| 677 | 1118 | 22115 | 1,869 | 0,0425 | 0,9575 | 0,16 | 1 |

**Appendix 1—table 14.** Pairs of lineage-defining sites of Alpha VOC (subset I) tend to have stronger signal of concordant evolution than the complementary subset of site pairs (subset II).

Site pairs were ordered by nominal p-values; for each of the two subsets, mean rank of site pairs included into that subset is provided. The difference between mean ranks (delta) obtained for the data is compared to the difference obtained for 400 samples from the null-model where substitutions in each site are independently and randomly distributed on the tree branches

(simulations). The probability that the difference of mean ranks for the data is greater than the difference of ranks for samples from the null-model is P{delta(simulations) ≤ delta(data)} <0.0025.

| | mean ranks for subset I (15 site pairs) | mean ranks for subset II (17005 site pairs) | delta |
|---|---|---|---|
| data | 234,7 | 8517,8 | –8283,1 |
| simulations | 2182,4 | 8516,1 | –6333,7 |

Number of lineage-defining sites: 6.
P{delta(simulations)≤delta(data)}<0.0025.

**Appendix 1—table 15.** Pairs of lineage-defining sites of Beta VOC tend to have stronger signal of concordant evolution than the complementary subset of site pairs.
See legend for *Appendix 1—table 14*.

| | mean ranks for subset I (15 site pairs) | mean ranks for subset II (17005 site pairs) | delta |
|---|---|---|---|
| data | 530,5 | 8517,5 | –7987,1 |
| simulations | 3021,4 | 8515,3 | –5494,0 |

Number of lineage-defining sites: 6.
P{delta(simulations)≤delta(data)}<0.0025.

**Appendix 1—table 16.** Pairs of lineage-defining sites of Delta VOC tend to have stronger signal of concordant evolution than the complementary subset of site pairs.
See legend for *Appendix 1—table 14*.

| | mean ranks for subset I (15 site pairs) | mean ranks for subset II (17005 site pairs) | delta |
|---|---|---|---|
| data | 3137,2 | 8515,2 | –5378,0 |
| simulations | 5545,9 | 8513,1 | –2967,2 |

Number of lineage-defining sites: 6.
P{delta(simulations)≤delta(data)}=0.005.

**Appendix 1—table 17.** Pairs of lineage-defining sites of Gamma VOC tend to have stronger signal of concordant evolution than the complementary subset of site pairs.
See legend for *Appendix 1—table 14*.

| | mean ranks for subset I (55 site pairs) | mean ranks for subset II (16965 site pairs) | delta |
|---|---|---|---|
| data | 543,3 | 8536,3 | –7993,0 |
| simulations | 3532,3 | 8526,6 | –4994,3 |

Number of lineage-defining sites: 11.
P{delta(simulations)≤delta(data)}<0.0025.

**Appendix 1—table 18.** Pairs of lineage-defining sites of Omicron VOC.
See legend for *Appendix 1—table 14*.

| | mean ranks for subset I (91 site pairs) | mean ranks for subset II (16929 site pairs) | delta |
|---|---|---|---|
| data | 5686,0 | 8525,7 | –2839,7 |
| simulations | 5540,6 | 8526,5 | –2985,9 |

Number of lineage-defining sites: 14.
P{delta(simulations)≤delta(data)}=0.635.

**Appendix 1—table 19.** Pairs of lineage-defining sites of Alpha VOC tend to have stronger signal of concordant evolution than the complementary subset of site pairs even if all Alpha and related lineages are excluded from the analysis.
See legend for *Appendix 1—table 14*.

|  | mean ranks for subset I (15 site pairs) | mean ranks for subset II (15385 site pairs) | delta |
|---|---|---|---|
| data | 316,8 | 7707,7 | –7390,9 |
| simulations | 4281,3 | 7703,8 | –3422,5 |

Number of lineage-defining sites: 6.
P{delta(simulations)≤delta(data)}<0.0025.

**Appendix 1—table 20.** No difference in the strength of concordant evolution is detected for pairs of lineage-defining sites of Beta VOC and the complementary subset of site pairs if all Beta and related lineages are excluded from the analysis.
See legend for *Appendix 1—table 14*.

|  | mean ranks for subset I (15 site pairs) | mean ranks for subset II (16275 site pairs) | delta |
|---|---|---|---|
| data | 4586,7 | 8148,8 | –3562,1 |
| simulations | 4522,3 | 8148,8 | –3626,6 |

Number of lineage-defining sites: 6.
P{delta(simulations)≤delta(data)}=0.5628.

**Appendix 1—table 21.** No difference in the strength of concordant evolution is detected for pairs of lineage-defining sites of Delta VOC and the complementary subset of site pairs if all Delta and related lineages are excluded from the analysis.
See legend for *Appendix 1—table 14*.

|  | mean ranks for subset I (15 site pairs) | mean ranks for subset II (16638 site pairs) | delta |
|---|---|---|---|
| data | 5934,7 | 8329,2 | –2394,5 |
| simulations | 5975,1 | 8329,1 | –2354,0 |

Number of lineage-defining sites: 6.
P{delta(simulations)≤delta(data)}=0.4462.

**Appendix 1—table 22.** Pairs of lineage defining sites of Gamma tend to have weaker signal of concordant evolution than the complementary subset of site pairs if all Gamma and related lineages are excluded from the analysis.
See legend for *Appendix 1—table 14*.

|  | mean ranks for subset I (45 site pairs) | mean ranks for subset II (16245 site pairs) | delta |
|---|---|---|---|
| data | 6424,4 | 8150,3 | –1725,8 |
| simulations | 5393,5 | 8153,1 | –2759,7 |

Number of lineage-defining sites: 10.
P{delta(simulations)≤delta(data)}=0.9728.
P{delta(simulations)≥delta(data)}=0.0272.
GISAID Identifier: EPI_SET_20220729hq doi:10.55876/gis8.220729hq.

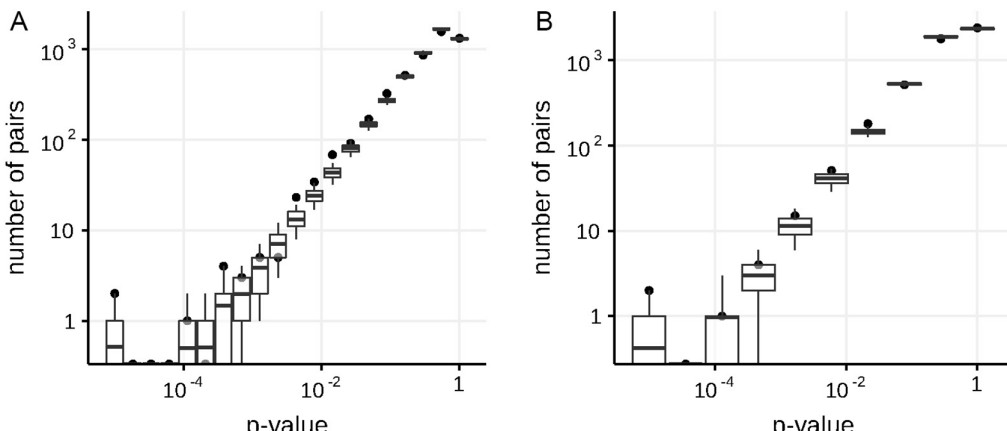

**Appendix 1—figure 1.** Numbers of predicted concordantly (**A**) and discordantly (**B**) evolving pairs for different nominal p-values in the simulated data in non-epistatic mode of evolution, compared to the null distribution for the ML phylogeny reconstructed by IQ-TREE. Black dots indicate numbers of site pairs having corresponded p-values in the simulated data, boxes with whiskers indicate distributions of numbers of site pairs having corresponded p-values among 400 samples from the null model (see Methods). Top and bottom of each box correspond to the 75th and 25th percentile, whiskers correspond to the 95th and 5th percentile. Vertical line corresponds to the 10% FDR threshold.

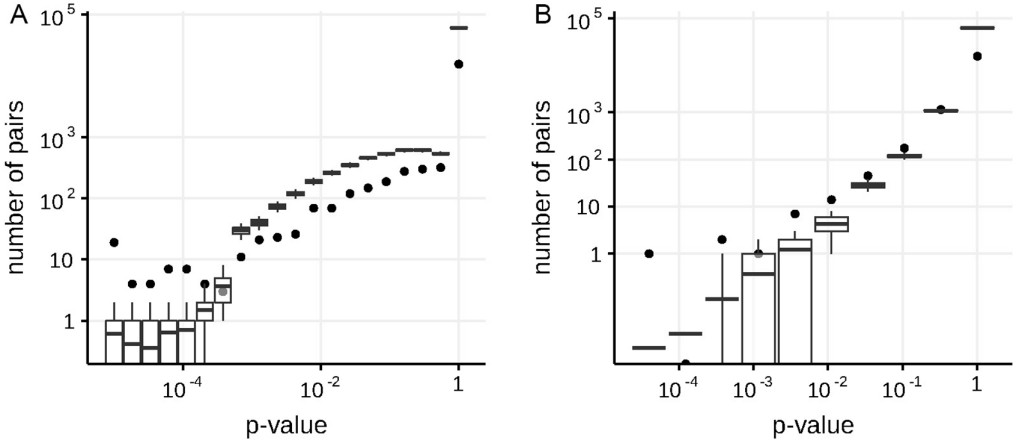

**Appendix 1—figure 2.** Numbers of predicted concordantly (**A**) and discordantly (**B**) evolving pairs for different nominal p-values in the S-gene of SARS-Cov-2, compared to the null distribution for the ML phylogeny reconstructed by IQ-TREE. See legend for the *Appendix 1—figure 1*.

