## [Editor Report]

Neverov and colleagues analyze patterns of correlated changes of amino acids in the SARS-CoV-2 spike protein to identify networks of interacting positions using an improved version of the previously validated method. Identifying such patterns of co-evolution is important for a better understanding of spike-protein evolution. The evidence for the identified co-evolving pairs is convincing, though the degree of certainty varies among the different identified groups of potentially interacting positions.

---

## [Decision Letter]

**Decision letter after peer review:**

Thank you for submitting your article "Evidence for coordinated evolution at amino acid sites of SARS-CoV-2 spike" for consideration by *eLife*. Your article has been reviewed by 3 peer reviewers, including Richard A Neher as the Reviewing Editor and Reviewer #1, and the evaluation has been overseen by Neil Ferguson as the Senior Editor.

Essential revisions:

All reviewers agreed that you have presented a sensible and intuitive investigation of epistasis in SARS-CoV-2 Spike evolution that uncovered or confirmed epistatically interacting networks of positions. This is a valuable addition to the literature. The main points raised during the review and the ensuing discussion are

1) Please make an effort to explain your method better. All reviewers struggled with some aspects (see reviews).

2) Please extend the discussion of limitations, for example, errors in phylogenetic reconstruction.

*Reviewer #1 (Recommendations for the authors):*

– Are these networks specific to SARS-CoV-2? Or even specific variants of SARS-CoV-2? Or are homologous sites also co-evolving in the broader group of sarbeco-viruses? Do these networks also feature in the work by Rodriguez-Rivas?

– We are currently seeing very rapid convergent evolution in multiple lineages (mutations at positions 346, 444, 460). How does convergent evolution driven by consistent selection pressure affect the specificity to detect epistatic groups? A more extensive discussion of this would be useful.

*Reviewer #2 (Recommendations for the authors):*

1. Although the method has been published before, I appreciated that the authors included a description here. But I found it a bit hard to follow in places. For example, I'm not sure why the authors chose the simulations they did to set the thresholds. I think the simulations mostly involved compensatory substitutions, but wouldn't SARS-CoV-2 epistatic evolution probably mostly involve positive substitutions that then potentiate new positive substitutions that were deleterious on the old background (so skirting valleys rather than crossing them)? I also wasn't sure what was going on with the simulation results, e.g., are the false positives clustered with the true positives or disjoint? What features distinguish true positives from false negatives? (Is it just how much they evolved?)

2. I'm not sure I totally understand the distinction between concordant and discordant evolution. I thought that it was concordant evolution when the derived alleles at both sites tended to be found in the same parts of the phylogeny and discordant when they tended to be on different branches, but the authors say that A653V and S982A occurring mostly on the D614G background is an example of discordant evolution.

3. The authors say that the epistasis inferences are robust to errors in the phylogeny, but the results actually seem fairly different between the two reconstructed phylogenies. Maybe reframe this?

4. I really appreciated Figures 3-5 (the example trees), which helped visualize what these signals actually look like. But I didn't totally understand them. Are all those pale blue and red dots included or excluded from the analysis? They look like they're on terminal branches, but they're unlikely to be errors, right?

5. I would be careful not to push the "enrichment VOC-defining mutations among the interacting sites" story too hard, given that they show that this is in large part because the data from the VOCs (except Omicron) is used to find the interactions

*Reviewer #3 (Recommendations for the authors):*

Comments and questions:

1. From paragraph at line 99 and Methods, it isn't clear – are concurrent changes (two mutations reconstructed to occur on the same branch) included in the analysis? (My instinct from the description is that they're not.) It seems like SARS-CoV-2 evolution might quickly fix epistatically coupled mutational pairs such that they appear coincidental on the tree (e.g. Omicron emergence, where we don't have mutational intermediates), and I would expect these to be an important source of signal for epistatic couplings.

2. For the simulation studies, is it clear why the revised method doesn't have false positives in the non-epistatic control simulations but does have some falsely identified pairs in the epistatic simulation?

3. Phylogeny figures: it would be very helpful if the main text or figure legend had more of a "narrative" description of how to interpret these figures besides the visual legend. Maybe I'm misinterpreting (it's a hard phylogeny to visually parse), but it seems like the dashed lines reflective of VOC annotation in Figure 3 are very far away from being monophyletic. This is surprising, it seems like at the very least a reliable phylogeny for this analysis needs to be getting those highest-level taxonomic groupings of VOCs correct. But, maybe I'm not interpreting the phylogeny correctly (which suggests some more care might need to be taken in visualization). Upon thinking further, it may be that because so many other sequences outside of Α are collapsed into the triangles, the polyatomic representation here is all of "Α" and "Α-like" things (that are just not annotated as Α?) surrounding the basal Α emergence and my initial impression is not as concerning as it appeared.

4. It would be interesting to add more context and speculation about the sets of sites that are seen to evolve concordantly and disconcordantly. For example, discussion and citations illustrate that cluster (I) of concordant sites are within an important class of antibody epitope. Double-check this, but does cluster (III) map to the NTD "antigenic supersite"? I also really liked the discussion of the signal of disconcordance between 501 and 675/677 in the Discussion. I think aspects of these types of interpretations could be included more proximal to the initial results themselves, too, to make it more clear to the reader what the relevance is of some of these pairs.

5. The observation in line 358 and Figure 5 made me jump to the reversion bias issue that is later discussed and clearly aware to the authors given discussion at line 374. I then came to understand that the many double mutations that appear on terminal branches on the phylogeny in Figure 5 are the light color indicating that they are excluded from analysis, which was my initial suggestion before I understood that these were already excluded. Rather than have a secondary color indicating excluded mutations that don't contribute to the epistatic pair discovery algorithm (assuming I understand correctly that terminal branches are not included in the algorithm), it might be more straightforward to just exclude these mutations from visualization entirely (and more clearly state that only internal branches are queried in the algorithm).

6. It is not entirely clear to me why the other epistatic mutations described in the cited paper on RBD mutations (Starr et al. 2022) are not seen in the phylogenetic signal. For example, in Figure 3E of that reference, substitutions are tabulated as singular occurrences on the phylogeny similar to the approach here, and suggests there is more signal between 501 and 449 than just the one 449H occurrence suggested in the discussion (e.g. mutations Y449H, D, N all contribute). Could this be related to point 1. above – are co-occurring substitutions being counted in the current algorithm, or is that potentially important signal being discarded? Or different sequence/phylogeny sources?

7. I want to thank the authors for continuing their important work and their outspoken statements during these difficult political times. We're all sending positive thoughts for peace in Ukraine.

---

## [Author Response]

Essential revisions:All reviewers agreed that you have presented a sensible and intuitive investigation of epistasis in SARS-CoV-2 Spike evolution that uncovered or confirmed epistatically interacting networks of positions. This is a valuable addition to the literature. The main points raised during the review and the ensuing discussion are1) Please make an effort to explain your method better. All reviewers struggled with some aspects (see reviews).2) Please extend the discussion of limitations, for example, errors in phylogenetic reconstruction.Reviewer #1 (Recommendations for the authors):– Are these networks specific to SARS-CoV-2? Or even specific variants of SARS-CoV-2? Or are homologous sites also co-evolving in the broader group of sarbeco-viruses? Do these networks also feature in the work by Rodriguez-Rivas?

In response to this comment, we contacted Dr. Rodriguez-Rivas who kindly provided the list of site pairs obtained in his analysis. Indeed, some of our concordantly evolving pairs were among the high DCA scoring pairs in the study of Rodriguez-Rivas, specifically, three pairs of sites in the RBD domain: (439, 441), (440, 442) and (441, 443) from cluster I; two pairs of sites in the N domain: (63, 64) and (69, 70) from cluster III; and one pair of sites in the S2_C domain: (1258, 1259). To test if this overlap is unexpected, for each Spike domain, we ordered pairs of sites located within that domain according to decreasing DCA scores and considered as “high scoring pairs” all pairs with ranks below or equal to the length of the corresponding domain: 305 for N, 178 for RBD and 40 for S2_C domain. For two domains, the intersections of the set of predicted concordantly evolving pairs with the set of DCA high scoring pairs were significantly larger than expected by chance alone: P=4.7e-4 (Fisher’s exact test) for RBD, P=3.2e-3 (Fisher’s exact test) for N domains. The third domain, S2_C, shows the same tendency with P=0.051 (Fisher’s exact test). There was only one concordantly evolving pair within this domain, which probably explains the relatively high p-value. (There were no pairs to consider in the remaining two domains, S1_C and S2). These findings indicate that all six sites of cluster I, four out of nine sites of cluster III and two sites in the S2_C domain which form a single pair cluster coevolve both at the larger timescales of sarbecovirus evolution and on the short timescale of two year evolution of SARS-Cov-2. To show these new results, we added a subsection entitled “Long-term coordinated evolution of Spike” to the Results section and expanded the Discussion section.

– We are currently seeing very rapid convergent evolution in multiple lineages (mutations at positions 346, 444, 460). How does convergent evolution driven by consistent selection pressure affect the specificity to detect epistatic groups? A more extensive discussion of this would be useful.

Indeed, as we discussed previously (Neverov et al., 2021), the fact that two sites evolve concordantly can mean (i) positive epistasis between them and/or (ii) fluctuating selection affecting them simultaneously. Our test does not distinguish between these two scenarios. As we also suggested previously (Neverov et al., 2021), they can be instead distinguished by patterns of distribution of non-consecutive mutations. Indeed, episodic selection simultaneously affecting two sites shortens phylogenetic distances between ALL mutations at these sites, including non-consecutive ones; by contrast, positive epistasis only leads to an excess of rapid consecutive mutations and does not bias distances between mutations occurring on different lineages.

To check whether coordinated evolution of some pairs can be explained by concordant episodic selection alone, we applied the test described in (Neverov et al., 2021), calculating the average distances between all pairs of non-consecutive substitutions for each concordantly evolving pair of sites. The results of this test are now included in the Results section (Tables 1, 2 and Appendix 1 – tables 10-13). Specifically, as pointed out by the Reviewer, site 444 has a strong signal of convergent evolution. For all concordantly evolving pairs formed by this site, as well as for other concordantly evolving pairs within cluster I (sites 439-444), non-consecutive mutations are in more closely related parts of the tree than expected, implying that their coordinated evolution indeed could be a result of coincident episodic selection. Another site mentioned by the Reviewer, 346, forms a concordantly evolving pair with site 12 on USHER phylogeny, but distances between non-consecutive mutations for this pair are not shorter than expected (z-score > 0) which suggests epistasis between these sites rather than concordant episodic selection. Site 460 is not involved in coordinated evolution, according to our test.

Overall, out of the 28 concordant site pairs that were detected on both phylogenies, for 12 pairs (10 of which were of cluster I) we could not rule out that their concordance stems from episodic selection alone, since their non-consecutive mutations also show signs of clustering (z-score < 0, p-value after Benjamini-Hochberg adjustment < 0.05). Concordant evolution of the other 16 pairs can hardly be explained without positive epistasis, since their non-consecutive mutations, in contrast to consecutive ones, either tend to avoid each other (pairs (69, 70) and (70, 144) from cluster III, z-score > 0, p-value after Benjamini-Hochberg adjustment < 0.05) or at least do not show any signs of clustering (for 11 pairs, z-score > 0; for the remaining 3 pairs, z-score < 0 but p-value of clustering even before adjustment is greater than 0.15).

A similar logic can be applied to discordantly evolving site pairs. For discordantly evolving pairs (614, 653) and (614, 982), non-consecutive mutations are phylogenetically more distant from each other than expected, suggesting that it also could be an example of discordant episodic selection rather than epistatic interaction. On the contrary, for discordant pairs (440, 681), (501, 675) and (501, 677), non-consecutive substitutions, in contrast to consecutive ones, tend to be closer to each other than expected. This can only happen as a result of negative epistasis, probably accompanied by concordant episodic selection. In summary, it appears that both epistasis and changes in selection between evolving lineages contribute to the observed signal.

We now include this analysis in the Results section and expand the Discussion, further elaborating on the mechanisms that could lead to concordant and discordant evolution.

Reviewer #2 (Recommendations for the authors):1. Although the method has been published before, I appreciated that the authors included a description here. But I found it a bit hard to follow in places. For example, I'm not sure why the authors chose the simulations they did to set the thresholds. I think the simulations mostly involved compensatory substitutions, but wouldn't SARS-CoV-2 epistatic evolution probably mostly involve positive substitutions that then potentiate new positive substitutions that were deleterious on the old background (so skirting valleys rather than crossing them)?

First, the FDR threshold was set based on the results of our test on the simulated dataset without epistasis, so the choice of the type of epistasis for simulations does not affect it. We only use epistatic simulations as a positive control, to illustrate that our method is capable of picking up epistasis. Second, to our knowledge, the type of epistatic interaction that you mention is not necessarily predominant in SARS-CoV-2. E.g., according to (Moulana et al., 2022), the majority of RBD mutations in BA.1 have a neutral or deleterious effect on ACE2 affinity on most genetic backgrounds, but together these mutations result in improvement in ACE2 affinity, which is an example of reciprocal sign epistasis simulated in our study.

Still, to ensure that our method is capable of detecting sign as well as reciprocal sign epistasis (i.e., skirting vs. crossing valleys), we now explicitly simulated non-reciprocal sign epistasis. These simulations were identical to the previous ones, except that for positive epistasis, the following values of fitness w for the four genotypes were used: w=0.9945 for the initial aa state, w=0.9955 and 0.8 for the two intermediate states aA and Aa, and the maximal w=1.0 for the AA state. Negative epistasis was simulated as previously. In this simulated dataset, our method still was able to detect positively epistatic pairs, although with a somewhat lower power: it detected 6 out of 10 positive epistatic pairs, and spuriously detected 14 other pairs as concordantly evolving. The power for detection of negative epistasis was slightly higher than in the original simulations; specifically, all 10 true pairs were inferred correctly, in addition to 8 spurious pairs. In summary, our approach is capable of detecting positive and negative epistasis independent of the exact model of epistasis.

I also wasn't sure what was going on with the simulation results, e.g., are the false positives clustered with the true positives or disjoint?

Indeed, false positives tend to cluster with true positives, although not pronouncedly (Author response image 1). Among the false positive pairs detected as concordantly evolving for higher FDR 30%, the pairs where one or both sites are involved in positive epistatic interactions are overrepresented, with the multinomial test p-value of 0.003 (Author response table 1). The same is true for spuriously detected discordantly evolving pairs: pairs where one or both sites are involved in negative epistatic interactions are overrepresented among them, with multinomial test p-value of 0.0018 (Author response table 2). See the answer to Reviewer 3 on question 2.

**Author response image 1. sa2fig1:** Site pairs detected as concordantly (I) or discordantly (II) evolving for FDR 30%, blue – site from a true concordantly evolving pair, red – site from a true discordantly evolving site pair, yellow – neutral; thick lines indicate true epistatic pairs.

**Author response table 1. sa2table1:** Observed frequencies of spurious detection of concordant evolution for different types of site pairs, compared to the expected (multinomial test p-value = 0). 0032). The number of possible FP pairs is calculated as the number of unordered combinations of sites that produce false positive pairs. For example, 20 sites in the simulation are involved in positive epistatic interactions and form 10 true epistatic pairs. The number of possible FP pairs can then be calculated as follows: Number of FP pairs = (Number of combinations of 2 sites from a set of 20) – (Number of true epistatic pairs) = 20*19/2 – 10 = 180.

**site types**	**#FP**	**#total possible FP**	**expected FP frequency**	**observed FP frequency**
**(pos,pos)**	7	180	0,0364	0,0843
**(pos,neg)**	7	400	0,0810	0,0843
**(neg,neg)**	0	190	0,0385	0
**(pos,neu)**	33	1200	0,2429	0,3976
**(neg,neu)**	6	1200	0,2429	0,0723
**(neu,neu)**	30	1770	0,3583	0,3614

**Author response table 2. sa2table2:** Observed frequencies of spurious detection of discordant evolution for different types of site pairs, compared to the expected. Multinomial test p-value = 0.0018.

**site types**	**#FP**	**#total possible FP**	**expected FP frequency**	**observed FP frequency**
**(pos,pos)**	0	190	0,0385	0
**(pos,neg)**	7	400	0,0810	0,1556
**(neg,neg)**	4	180	0,0364	0,0889
**(pos,neu)**	5	1200	0,2429	0,1111
**(neg,neu)**	19	1200	0,2429	0,4222
**(neu,neu)**	10	1770	0,3583	0,2222

What features distinguish true positives from false negatives? (Is it just how much they evolved?)

Our test is heuristic, and we do not expect it to detect epistasis perfectly. Our power depends on several factors. First, our ability to infer the presence and direction of epistasis will depend on the exact shape of the fitness landscape: it will be higher for the reciprocal sign epistasis, and lower for other types of epistasis. Furthermore, the identity of the ancestral alleles matters. In our simulations, we assume that the ancestral state corresponds to the fittest allele pair. In reality, in some cases, the most recent ancestor of the considered sample may carry the state that is not the most fit, or not uniquely the most fit; if so, the sign of the epistasis (positive vs. negative) will depend on the ancestral state. In general, in case of epistasis, the difference between concordant and discordant evolution is methodological rather than inherent in the nature of the underlying evolutionary process: we detect an excess of consecutive substitutions in case of concordant evolution and a deficiency of consecutive substitutions in case of discordant evolution, but both indicate that an allele at one site preferentially arises on the background of some, but not others, alleles at another site.

In our main analysis, we do not observe any false negatives in simulations of positive epistasis, but we do get some false negatives in simulations of negative epistasis. In our negative epistatic simulations, we assume a fitness ridge rather than a fitness valley. It appears that at these site pairs, a substitution into one of the two variants at the “end of the ridge” (aA or Aa) happens early on, and in subsequent evolution, the signal of negative epistasis is masked by the signal of positive epistasis.

2. I'm not sure I totally understand the distinction between concordant and discordant evolution. I thought that it was concordant evolution when the derived alleles at both sites tended to be found in the same parts of the phylogeny and discordant when they tended to be on different branches, but the authors say that A653V and S982A occurring mostly on the D614G background is an example of discordant evolution.

Yes, this is correct: we infer concordant evolution when the derived alleles at both sites tend to be found in the same parts of the phylogeny, and discordant when they tend to be on different branches. We do not claim that sites 653 and 982 evolve discordantly. Rather, sites 653 and 614 evolve discordantly, as well as sites 982 and 614. This signal comes from the fact that both A653V and S982A avoid occurring on the background of reversion to D at site 614. We now reworded this sentence for clarity.

3. The authors say that the epistasis inferences are robust to errors in the phylogeny, but the results actually seem fairly different between the two reconstructed phylogenies. Maybe reframe this?

We now mitigate this statement in the Discussion, rephrasing it as follows: “The choice of the method of phylogenetic reconstruction affected the set of predicted pairs (tables 1, S7 and S8); however, over 60% of concordantly evolving pairs detected using the IQ-TREE phylogeny were also detected using the UShER tree”.

4. I really appreciated Figures 3-5 (the example trees), which helped visualize what these signals actually look like. But I didn't totally understand them. Are all those pale blue and red dots included or excluded from the analysis? They look like they're on terminal branches, but they're unlikely to be errors, right?

Thank you. Yes, pale dots only represent mutations on terminal branches, which were excluded from the analysis. For clarity of presentation, the mutations that are excluded from the analysis are now also excluded from the visualization in the main text (Figures 3 and 5). Visualizations showing all mutations, including those on terminal branches, are moved to Figure 3—figure supplement 1 and Figure 5—figure supplement 1. We also expanded the descriptions for Figures 3 and 5.

5. I would be careful not to push the "enrichment VOC-defining mutations among the interacting sites" story too hard, given that they show that this is in large part because the data from the VOCs (except Omicron) is used to find the interactions

We agree, and now paraphrase lines 19-21 in the Abstract accordingly.

Reviewer #3 (Recommendations for the authors):Comments and questions:1. From paragraph at line 99 and Methods, it isn't clear – are concurrent changes (two mutations reconstructed to occur on the same branch) included in the analysis? (My instinct from the description is that they're not.)

No, we considered all consequent mutation pairs including those on the same branches. Now we clarified this earlier in the Methods.

It seems like SARS-CoV-2 evolution might quickly fix epistatically coupled mutational pairs such that they appear coincidental on the tree (e.g. Omicron emergence, where we don't have mutational intermediates), and I would expect these to be an important source of signal for epistatic couplings.

Yes, we fully agree. Such pairs do actually contribute to our findings, e.g., they contribute to the signal of positive epistasis between sites 484 and 655 (purple diamonds in Figure 5).

2. For the simulation studies, is it clear why the revised method doesn't have false positives in the non-epistatic control simulations but does have some falsely identified pairs in the epistatic simulation?

One reason may be the following. The fact that a site is involved in an epistatic interaction (with any other site) biases the phylogenetic distribution of replacements at it away from random; e.g., trailing mutations will be biased towards branch tips. Such a bias may then lead to a spurious signal of epistasis between this site and a neutral site, leading to false positives in our simulation. Indeed, many of the spurious epistatic pairs are between a neutral site and an epistatic site (see response to comment 1 of Reviewer 2).

3. Phylogeny figures: it would be very helpful if the main text or figure legend had more of a "narrative" description of how to interpret these figures besides the visual legend.

We now expanded the descriptions for Figures 3 and 5.

Maybe I'm misinterpreting (it's a hard phylogeny to visually parse), but it seems like the dashed lines reflective of VOC annotation in Figure 3 are very far away from being monophyletic. This is surprising, it seems like at the very least a reliable phylogeny for this analysis needs to be getting those highest-level taxonomic groupings of VOCs correct. But, maybe I'm not interpreting the phylogeny correctly (which suggests some more care might need to be taken in visualization). Upon thinking further, it may be that because so many other sequences outside of Α are collapsed into the triangles, the polyatomic representation here is all of "Α" and "Α-like" things (that are just not annotated as Α?) surrounding the basal Α emergence and my initial impression is not as concerning as it appeared.

First, we note that we use VOC annotation for just one purpose: to exclude the samples pertaining to a VOC from the analysis of the lineage defining mutations of this VOC. This affects which lineages are included in the analysis, but does not affect our object of study – the distribution of mutations in the remaining lineages.

Second, indeed, in our tree, the Α VOC is not monophyletic (Author response image 2). We now researched why this is the case. In our study, we relied on PANGOLIN annotations provided in GISAID metadata for VOC classification. It seems that sequences outside the phylogenetic clades corresponding to VOCs were ascribed to wrong categories by PANGOLIN. Instability of lineage assignments by PANGOLIN has been described previously (https://virological.org/t/sars-cov-2-lineage-assignment-is-more-stable-with-usher/781). To study this further, we ran the latest version of Pangolin (v. 4.1.3) both in the fast mode (when pangoLEARN is used for classification) and the default mode (when subtypes are assigned by placing the sample on the current publicly available phylogenetic tree using USHER), and considered PANGOLIN’s QC statistics (https://cov-lineages.org/resources/pangolin/output.html). One useful statistic is “conflict” which is greater than zero if a sequence could be assigned into multiple categories; another statistic, “ambiguity_score”, reflects the portion of sites used for classification that were imputed to the reference values. We compared “conflict” and “ambiguity_score” statistics for two sets of samples classified as VOCs: those contained in the largest clade of this VOC (below called TP from “true pangolin lineages”), and the remaining sequences ascribed to the same VOC lineages (below called FP from “false pangolin lineages”).

**Author response image 2. sa2fig2:** Phylogenetic clades annotated as VOCs according to the GISAID metadata as of 07. 09.2021 (I), and to the newer PANGOLIN 4.1.3, used in the default mode (II). Α is shown in red color, Β in blue, Γ in green, and Δ in orange. There are more red (α) branches outside the main α clade in (I) than in (II).

In both modes, FP sequences had significantly higher “conflict” scores than TP sequences (Mann Whitney U-test p-value = 1e-25 and 1e-11 for fast and default modes respectively), indicating that the classification of FP sequences is ambiguous. While SARS-CoV-2 lineage assignment is more stable with UShER compared to the pangoLEARN mode (https://virological.org/t/sars-cov-2-lineage-assignment-is-more-stable-with-usher/781), some polyphyly is also observed in the UShER-based annotation (Author response image 2).

The analysis in our text is based on the annotation in GISAID metadata as of 07.09.2021, which in turn is apparently based on pangoLEARN. We chose to retain it for the following reason. The two approaches only differ in which sequences are excluded from analysis. As under the GISAID annotation, more sequences belonging to the VOC are left outside the VOC’s main clade, exclusion based on this annotation is more conservative: it ensures that even those sequences whose inclusion in the VOC is ambiguous are not included in the analysis of concordance. The fact that our approach is conservative is now explained.

4. It would be interesting to add more context and speculation about the sets of sites that are seen to evolve concordantly and disconcordantly. For example, discussion and citations illustrate that cluster (I) of concordant sites are within an important class of antibody epitope.

Thank you for this suggestion. Indeed, mutations at sites 439-444 from the cluster (I) affect neutralization by monoclonal and polyclonal antibodies (Barnes et al., 2020; Harvey et al., 2021). We added this to Results.

Double-check this, but does cluster (III) map to the NTD "antigenic supersite"?

Five out of nine sites from the cluster (III) are located in the NTD hypervariable loops close to their flanks: sites 67, 69 and 70 are within the loop N2 (positions 67-81), site 144 is located within the loop N3 (positions 140-158), and sites 259 and 261 are within the loop N5 (positions 241-263). It was shown that changes of lengths and sequences of these loops mediate Spike membrane fusion, cell entry and extracellular stability (Cantoni et al., 2022; Qing et al., 2021). Loops N1 (positions 14-26), N3 and N5 contribute to NTD supersite (Cerutti et al., 2021; McCallum et al., 2021). In the loop N1 there are three sites (18, 20 and 26) from the cluster (IV). Also, the concordantly evolving site pair (155,157) comprises sites flanking the N3 loop. Yet another concordantly evolving pair (13, 152) comprises a site from the signal peptide and a site from the N3 loop. It has been previously shown that mutations at some sites of the signal peptide could abrogate virus neutralization by antibodies due to changes of the signal peptide cleavage site and therefore changes of the structure of NTD, e.g. P9SQL and S12P (McCallum et al., 2021). However, among the concordantly evolving sites in NTD identified in our study, only sites 18 and 144 carry Ab escape mutations. We now added this to the Results.

I also really liked the discussion of the signal of disconcordance between 501 and 675/677 in the Discussion. I think aspects of these types of interpretations could be included more proximal to the initial results themselves, too, to make it more clear to the reader what the relevance is of some of these pairs.

Thank you. Text rearranged as suggested.

5. The observation in line 358 and Figure 5 made me jump to the reversion bias issue that is later discussed and clearly aware to the authors given discussion at line 374. I then came to understand that the many double mutations that appear on terminal branches on the phylogeny in Figure 5 are the light color indicating that they are excluded from analysis, which was my initial suggestion before I understood that these were already excluded. Rather than have a secondary color indicating excluded mutations that don't contribute to the epistatic pair discovery algorithm (assuming I understand correctly that terminal branches are not included in the algorithm), it might be more straightforward to just exclude these mutations from visualization entirely (and more clearly state that only internal branches are queried in the algorithm).

Thank you for the suggestion. Indeed, to infer coordinated evolution of sites, we considered only the mutations which occurred on the internal branches. In the revision, we now exclude the mutations that are excluded from the analysis from the visualization in the main text (Figures 3 and 5). Visualizations showing all mutations, including those on terminal branches, are moved to Figure 3—figure supplement 1 and Figure 5—figure supplement 1.

6. It is not entirely clear to me why the other epistatic mutations described in the cited paper on RBD mutations (Starr et al. 2022) are not seen in the phylogenetic signal. For example, in Figure 3E of that reference, substitutions are tabulated as singular occurrences on the phylogeny similar to the approach here, and suggests there is more signal between 501 and 449 than just the one 449H occurrence suggested in the discussion (e.g. mutations Y449H, D, N all contribute). Could this be related to point 1. above – are co-occurring substitutions being counted in the current algorithm, or is that potentially important signal being discarded? Or different sequence/phylogeny sources?

Four changes at site 449 occurred on branches of our phylogenetic tree: three changes Y449H and one Y449F. Y449F occurred on the background of N at site 501, and all three changes Y449H occurred on the background of Y at site 501. One of the Y449H changes coincided with N501Y on a single internal branch which did not however result in a prolific clade (it carried just two samples), and the remaining Y449H and Y449F changes occurred on terminal branches, consistent with the hypothesis that they are deleterious. Thus, our data agree with the observation in (Starr et al., 2022) that Y449H occurs mainly on the background of allele Y and not N at site 501, but that Y449H and Y449F are deleterious in both contexts at site 501 (see figures 2C and 3D in their paper).

The fact that we do not observe a signal of concordance between sites 501 and 449 is due to two facts: first, we used less data, and therefore had less power; second, for the sake of caution, we discarded terminal branches which are more prone to artifacts (see Methods). Starr et al. used the global SARS-CoV-2 phylogeny including all genome sequences available in GISAID on 25.05.2022, which is much more data than was used in our phylogenetic analysis. This allowed them to observe a sufficient number of occurrences of the deleterious mutation Y449H on internal branches to estimate the excess of occurrences on the background of 501Y. However, the excess of occurrence of Y449H on the background of 501Y shown in figure 3E in the Starr et al. paper is moderate, and its p-value – the probability to be observed just by chance alone – has not been estimated. For our tree, the p-value of concordance for this pair was 8e-4 if we considered mutations occurring both on internal and terminal branches, but the corresponding FDR was relatively high (30%). So, the observed pattern of mutations in these sites is significantly different from that expected under the assumption of independent evolution, but the fraction of false positives is high for the corresponding significance level. We thus can conclude that our observations generally agree with that of Starr et al.

References

Barnes CO, Jette CA, Abernathy ME, Dam K-MA, Esswein SR, Gristick HB, Malyutin AG, Sharaf NG, Huey-Tubman KE, Lee YE, Robbiani DF, Nussenzweig MC, West AP, Bjorkman PJ. 2020. SARS-CoV-2 neutralizing antibody structures inform therapeutic strategies. *Nature* 588:682–687. doi:10.1038/s41586-020-2852-1

Cantoni D, Murray MJ, Kalemera MD, Dicken SJ, Stejskal L, Brown G, Lytras S, Coey JD, McKenna J, Bridgett S, Simpson D, Fairley D, Thorne LG, Reuschl A-K, Forrest C, Ganeshalingham M, Muir L, Palor M, Jarvis L, Willett B, Power UF, McCoy LE, Jolly C, Towers GJ, Doores KJ, Robertson DL, Shepherd AJ, Reeves MB, Bamford CGG, Grove J. 2022. Evolutionary remodelling of N-terminal domain loops fine-tunes SARS-CoV-2 spike. *EMBO Rep* 23:e54322. doi:10.15252/embr.202154322

Cerutti G, Guo Y, Zhou T, Gorman J, Lee M, Rapp M, Reddem ER, Yu J, Bahna F, Bimela J, Huang Y, Katsamba PS, Liu L, Nair MS, Rawi R, Olia AS, Wang P, Zhang B, Chuang G-Y, Ho DD, Sheng Z, Kwong PD, Shapiro L. 2021. Potent SARS-CoV-2 neutralizing antibodies directed against spike N-terminal domain target a single supersite. *Cell Host Microbe* 29:819-833.e7. doi:10.1016/j.chom.2021.03.005

Harvey WT, Carabelli AM, Jackson B, Gupta RK, Thomson EC, Harrison EM, Ludden C, Reeve R, Rambaut A, Peacock SJ, Robertson DL. 2021. SARS-CoV-2 variants, spike mutations and immune escape. *Nat Rev Microbiol* 19:409–424. doi:10.1038/s41579-021-00573-0

McCallum M, De Marco A, Lempp FA, Tortorici MA, Pinto D, Walls AC, Beltramello M, Chen A, Liu Z, Zatta F, Zepeda S, di Iulio J, Bowen JE, Montiel-Ruiz M, Zhou J, Rosen LE, Bianchi S, Guarino B, Fregni CS, Abdelnabi R, Foo S-YC, Rothlauf PW, Bloyet L-M, Benigni F, Cameroni E, Neyts J, Riva A, Snell G, Telenti A, Whelan SPJ, Virgin HW, Corti D, Pizzuto MS, Veesler D. 2021. N-terminal domain antigenic mapping reveals a site of vulnerability for SARS-CoV-2. *Cell* 184:2332-2347.e16. doi:10.1016/j.cell.2021.03.028

Moulana A, Dupic T, Phillips AM, Chang J, Nieves S, Roffler AA, Greaney AJ, Starr TN, Bloom JD, Desai MM. 2022. Compensatory epistasis maintains ACE2 affinity in SARS-CoV-2 Omicron BA.1. *Nat Commun* 13:7011. doi:10.1038/s41467-022-34506-z

Neverov AD, Popova AV, Fedonin GG, Cheremukhin EA, Klink GV, Bazykin GA. 2021. Episodic evolution of coadapted sets of amino acid sites in mitochondrial proteins. *PLOS Genet* 17:e1008711. doi:10.1371/journal.pgen.1008711

Qing E, Kicmal T, Kumar B, Hawkins GM, Timm E, Perlman S, Gallagher T. 2021. Dynamics of SARS-CoV-2 Spike Proteins in Cell Entry: Control Elements in the Amino-Terminal Domains. *mBio* 12:e01590-21. doi:10.1128/mBio.01590-21

Starr TN, Greaney AJ, Hannon WW, Loes AN, Hauser K, Dillen JR, Ferri E, Farrell AG, Dadonaite B, McCallum M, Matreyek KA, Corti D, Veesler D, Snell G, Bloom JD. 2022. Shifting mutational constraints in the SARS-CoV-2 receptor-binding domain during viral evolution. *Science* 377:420–424. doi:10.1126/science.abo7896